# Automatically tracking feeding behavior in populations of foraging *C. elegans*

**Elsa Bonnard[1†], Jun Liu[1†], Nicolina Zjacic[1,2], Luis Alvarez[1], Monika Scholz[1]\***

[1]Max Planck Research Group Neural Information Flow, Max Planck Institute for Neurobiology of Behavior – caesar, Bonn, Germany; [2]Institute of Medical Genetics, University of Zurich, Zurich, Switzerland

**Abstract** *Caenorhabditis elegans* feeds on bacteria and other small microorganisms which it ingests using its pharynx, a neuromuscular pump. Currently, measuring feeding behavior requires tracking a single animal, indirectly estimating food intake from population-level metrics, or using restrained animals. To enable large throughput feeding measurements of unrestrained, crawling worms on agarose plates at a single worm resolution, we developed an imaging protocol and a complementary image analysis tool called PharaGlow. We image up to 50 unrestrained crawling worms simultaneously and extract locomotion and feeding behaviors. We demonstrate the tool's robustness and high-throughput capabilities by measuring feeding in different use-case scenarios, such as through development, with genetic and chemical perturbations that result in faster and slower pumping, and in the presence or absence of food. Finally, we demonstrate that our tool is capable of long-term imaging by showing behavioral dynamics of mating animals and worms with different genetic backgrounds. The low-resolution fluorescence microscopes required are readily available in *C. elegans* laboratories, and in combination with our python-based analysis workflow makes this methodology easily accessible. PharaGlow therefore enables the observation and analysis of the temporal dynamics of feeding and locomotory behaviors with high-throughput and precision in a user-friendly system.

**\*For correspondence:**
monika.scholz@mpinb.mpg.de

[†]These authors contributed equally to this work

**Competing interest:** The authors declare that no competing interests exist.

## Editor's evaluation

In this study, Bonnard and colleagues report a new method to assay feeding rates in *C. elegans*. Imaging fluorescence in the pharynx with subsequent image processing steps they make it possible to record pharyngeal pumping across freely behaving animal populations over periods up to 3 hs. They validate their method in different behavioural paradigms and with various feeding mutants.

## Introduction

Feeding is important for animal physiology, affecting energy balance, longevity, healthspan, or aging (*Fontana and Partridge, 2015*; *Trepanowski et al., 2011*; *Balasubramanian et al., 2017*). Accurate measurements of feeding behavior are required to assess these physiological effects. Thanks to its fully sequenced and annotated genome that shares at least 50% homology with human genome and the availability of advanced genome editing tools, short life cycle and transparency, the roundworm *Caenorhabditis elegans* is a powerful model to study feeding. Research in *C. elegans* has shed light on how internal states such as hunger, peptidergic, and bioaminergic regulation (*Avery and Horvitz, 1990*; *You et al., 2006*; *You et al., 2008*; *Lee et al., 2017*; *Song and Avery, 2012*; *Scholz et al., 2017*; *Kang and Avery, 2021*; *Srinivasan et al., 2008*; *Hobson et al., 2006*), and decision making affect feeding (*Katzen et al., 1983*; *Shtonda and Avery, 2006*). Being able to detect feeding in large populations at single-animal resolution would enable further insight into inter-animal variability,

**eLife digest** A small worm called *C. elegans* is constantly hungry. It spends all its time looking for food or eating. Hunger and environmental factors, like light, influence its feeding behavior. Studying these worms has helped scientists learn how feeding affects health, longevity, and aging. Feeding studies might also help scientists learn how the nervous system works and how it controls feeding.

Most studies have used one of two approaches. Scientists may measure how much food a group of *C. elegans* eat by measuring food before and after it is offered to the worms. Or they restrain individual worms and measure the movement of a tube-like muscle, called the pharynx, which the animals use to vacuum up food. Restraining the worms can alter their behavior or brain activity, and studying group feeding habits may miss individual differences, so neither is optimal. Ideally, scientists could measure the feeding activity of many free-ranging worms, but because the movements of the pharynx are small, that too can be a challenge.

Bonnard, Liu et al. developed a software tool that automatically detects and measures feeding behavior in a group of about 30 free-ranging *C. elegans* simultaneously. In the experiments, Bonnard, Liu et al. genetically engineered worms expressing a fluorescent protein in their pharynx, making it possible to measure its movements with a microscope. They used the microscope to capture images of 30-50 animals at a time as they foraged for food in a dish. Then, they used the software to analyze the data they collected. Over three days and five imaging sessions, Bonnard and Liu et al. tracked the feeding behavior of about 1,000 animals under different conditions. The experiments show that the pharynx grows rapidly during early worm development when the worms quadruple their length, but the rate of pharynx muscle contractions stays the same. They also showed the technique could measure feeding behaviors in animals with different genetic backgrounds, ages, or those engaged in behaviors like mating.

The tool allows for larger and longer-term studies of worm feeding behaviors than previous approaches. Bonnard, Liu et al. made their software, called PharaGlow, available for use by other researchers. The tool may make feeding measurements a routine part of *C. elegans* studies. It will allow scientists to gain new insights into the role of feeding in a range of processes, including aging, fitness, mating, and overall health. Follow-up studies could determine if these findings are general strategies that also apply to other animals.

internal states and subtle modulatory effects in the temporal dynamics of feeding. To understand the coupling of multiple behaviors, such as locomotion and feeding, it is required to allow animals to roam freely while feeding and assess both behaviors at the same time. Here, we propose a method to measure the feeding activity in unrestrained populations of *C. elegans* with sufficient temporal resolution to observe single feeding events.

*C. elegans* feeds on bacteria and other small microorganisms by drawing in a suspension of food particles from the environment. The bacteria are ingested and separated from the liquid by the pumping action of its powerful pharyngeal muscles (*Seymour et al., 1983*; *Avery and Shtonda, 2003*; *Fang-Yen et al., 2009*). Transport of the bacteria proceeds with occasional peristaltic contractions that move food further toward the terminal bulb where a hard cuticular structure, the grinder, crushes the bacteria before they are pushed into the intestine (*Albertson and Thomson, 1976*). Pumping is the limiting step for food intake that is, the total food consumed is the product of pumping rate and external food concentration (*Seymour et al., 1983*; *Avery and Shtonda, 2003*; *Fang-Yen et al., 2009*). Pumping is inherently a stochastic process (*Lee et al., 2017*). It has been suggested that stochastic pumping results from a decision making process that serves to regulate pumping based on food availability (*Scholz et al., 2017*). Even in the absence of food, pumping has been observed and interpreted as a mechanism for food sampling (*Lee et al., 2017*; *Scholz et al., 2017*; *Trojanowski et al., 2016*). On average, pumping occurs up to 300 times per minute when food is abundant (*Lee et al., 2017*; *Song and Avery, 2012*; *Scholz et al., 2016*). Pumping rates are altered in response to the type, concentration, size, and familiarity of the surrounding bacteria (*Lee et al., 2017*; *Avery and Shtonda, 2003*; *Scholz et al., 2016*; *Song et al., 2013*). The behavioral and metabolic context, such as hunger, satiety, and mating drive also influence the rate of food intake (*Avery and Horvitz, 1990*; *You et al., 2006*; *You et al., 2008*; *Gruninger et al., 2006*). Feeding behavior is thus regulated

**Table 1.** Comparison of methods for measuring pumping.

| Technique | Single pump | Single worm | Animals/ setup | Method | Label | Constrained | Source |
|---|---|---|---|---|---|---|---|
| Bioluminescent bacteria | No | No | 100–1000 | Microscopy | No | No | *Ding et al., 2020* |
| Luciferase expressing worms | No | Yes | 100 | Microscopy | Yes | No | *Rodríguez-Palero et al., 2018* |
| Optical density | No | No | 100–1000 | Absorption | No | No | *Gomez-Amaro et al., 2020* |
| Tracking microscope | Yes | Yes | 1 | Microscopy | No | No | *Li et al., 2012*; *Cermak et al., 2020*; *Zou et al., 2018* |
| pWarp | Yes | Yes | 4 | Microscopy | No | microfluidic | *Scholz et al., 2016* |
| NemaChip | Yes | Yes | 8 | Electrophysiology / EPG | No | microfluidic | *Lockery et al., 2012* |
| Manual counting | Yes | Yes | 1 | Microscopy | No | No | *Song and Avery, 2012*; *Dallière et al., 2016*; *Bhatla et al., 2015* and many others |
| PharaGlow | **Yes** | **Yes** | **1–50** | **Microscopy** | **Yes** | **No** | This work |

at different time scales ranging from immediate neuro-muscular activity (*Trojanowski et al., 2016*; *Raizen et al., 1995*; *McKay et al., 2004*; *Raizen and Avery, 1994*), to the intermediate scales of food choice and foraging (*Scholz et al., 2017*; *Katzen et al., 1983*; *Li et al., 2012*), to longer-term life history traits and behavioral state changes of the animal (*Avery and Horvitz, 1990*; *Cermak et al., 2020*).

Because of the transparent body of *C. elegans*, the pharynx can be directly observed through light microscopy, which in principle enables simultaneous detection of food particles (bacteria), muscular motion, and locomotion (*Fang-Yen et al., 2009*). However, these experiments are often performed in immobilized animals, which can introduce artifacts in the observed behavior, as the activity of the body wall muscles feedbacks to the pharynx via parallel synaptic and hormonal routes (*Takahashi and Takagi, 2017*; *Izquierdo et al., 2022*). While desirable, imaging feeding in unrestrained animals, especially in large populations, is challenging due to the disparate time- and length scales of the motions involved. While worms move over centimeters within minutes (*Ramot et al., 2008*; *Swierczek et al., 2011*), the observable pharyngeal contractions are over μm within ms (*Fang-Yen et al., 2009*), making large-scale foraging experiments technically challenging.

Existing techniques to measure feeding fall broadly into two categories. The first focuses on indirect measures of population food intake, and the second detects each pumping contraction (*Table 1*). Indirect food intake measures rely either on labeling the food intake of the worm, for example using bioluminescent bacteria (*Ding et al., 2020*), fluorescent bacteria (*You et al., 2008*; *Andersen et al., 2014*), or fluorescent beads (*Fang-Yen et al., 2009*; *Kiyama et al., 2020*), or by measuring the remaining food concentration over time in large liquid cultures of worms (*Gomez-Amaro et al., 2020*). However, liquid culture neither allows direct measure of pumping activity nor of feeding related behavior such as locomotion toward food. Resolving single pump information can be achieved by combining bright-field microscopy with live worm tracking to remove center of mass motion and enable imaging of the grinder (*Li et al., 2012*; *Cermak et al., 2020*; *Zou et al., 2018*), or alternatively by constraining animals in microfluidics. In tracking and constrained configurations, one can read out pumps by directly following the grinder motion in the pharynx (*Lee et al., 2017*; *Scholz et al., 2016*). A complementary technique relies on recording electropharyngeograms that detect the signature of muscular contractions in a small population of constrained animals without requiring a tracking microscope (*Lockery et al., 2012*). Despite these numerous approaches, what is lacking is a method that allows time-resolved pumping detection in large populations of unrestrained crawling animals.

We wanted to fill the gap to allow imaging of pumping activity at high-throughput with single-pump temporal resolution in unrestrained animals, while using only optical setups already available in most *C. elegans* laboratories. Our method is based on epi-fluorescence microscopy of the pharyngeal muscle with a cost effective, large chip camera that enables imaging of many worms as they explore freely on an agarose plate. We determined that the method is relatively insensitive to the optical

instrument used, and does not require high-end or custom optics. The accompanying analysis software (PharaGlow) is written in Python and can be accessed using beginner friendly semi-graphical jupyter notebooks. PharaGlow is available under a permissive open source license. We demonstrate the usability and throughput of the method for multiple use cases, including those not previously possible in restrained animals, such as repeated imaging of a population of developing animals and investigating the coupling of locomotion and pumping rates. Finally, we demonstrate the utility of our approach by imaging a pair of mating animals over more than 1 hr.

## Results

### Detection of pumping rates in crawling animals

To enable automated, high-throughput detection of pumping in animals crawling on culture plates, we combined epi-fluorescence microscopy with a large area scan camera (*Figure 1A*). Typically, pumping is detected by manual or automated counting using high magnification to resolve the motion of the grinder in the terminal bulb (*Dallière et al., 2016*; *Bhatla et al., 2015*). Using animals expressing a fluorescent protein in the pharyngeal muscle allowed us to image at a lower magnification compared to the resolution required for bright-field imaging of the grinder. Specifically, we image animals expressing YFP under the pharyngeal myosin promoter *myo-2p* (*gnaIs1 [myo-2p::yfp]*), which is present in all pharyngeal muscles except pm1 and pm2 (*Okkema et al., 1993*; *Okkema and Fire, 1994*).

By using a low magnification of 1x, we could image a field of view of 7 by 5 mm, corresponding to multiple body lengths of the worms (*Figure 1A*). We simultaneously imaged tens of animals (typically 30–50) as they crawled and analyzed their behavior off-line using our custom analysis software (*Figure 1B*; *Video 1*). The analysis pipeline combines a particle-tracking workflow with custom shape segmentation of the fluorescent pharynx (*Figure 1C*). After detecting and tracking the pharynges in the field of view, the contour and centerlines are fitted. The centerline and width are used to virtually straighten the animal (*Figure 1D*). We then extract a pump metric from the straightened images based on the standard deviation of the fluorescence along the dorso-ventral axis (DV-axis) of the animal, which reflects pumping events (*Figure 1E and F*). By averaging images during these putative detected pumping events, we determined that this metric is sensitive to the opening of the pharyngeal lumen and contraction of the terminal bulb and thus indeed corresponds to pumping events (*Figure 1G*).

Although the low magnification (1x) we use to image the animals allows us to increase the number of observed animals, this could compromise pumping detection. To determine how accurate our software detects pumping in these imaging conditions, we compared the results of our automated method and a manual annotator. Since manual annotation of pumping rates is still widely used, but practiced at higher magnifications, we simultaneously imaged worms at a magnification of 10x (pixel size 240 nm/px) in bright field and fluorescence (*Figure 1H*; *Video 2*). A human expert counted pumps in the videos acquired using the bright-field channel. We then ran our automated analysis on the video acquired on the fluorescence channel, but downscaled to 1x (pixel size 2.4 µm/px). We found that PharaGlow was able to accurately detect pumping in these videos, and the resulting rate and counts were in agreement with the human expert. Both methods result in a comparable mean pumping rate for the animals counted (*Figure 1I*), with a deviation between the human and automated results of less than 2 pumps per 10 s (*Figure 1J*). To score a typical experiment of 30 animals over 5 min of recording time, the human experimenter would need, at best, to count for at least 150 min of data (real time). This time is regularly longer, as accurate counting often requires scrutinizing the recordings in slow motion or visualizing the same part of the recording several times. PharaGlow is therefore able to automatically and reliably detect pumping in low-resolution, large field of view images, enhancing the number of animals which can be scored simultaneously.

### Developmental pumping

Having developed this new high-throughput method which enables accurate measurements of many animals simultaneously, we wondered how pumping changes over the course of development, where the animal changes its size and its energy needs. During development, the pharynx grows with the body (*Knight et al., 2002*), but the ratio between pharynx and body length decreases from L1 stage

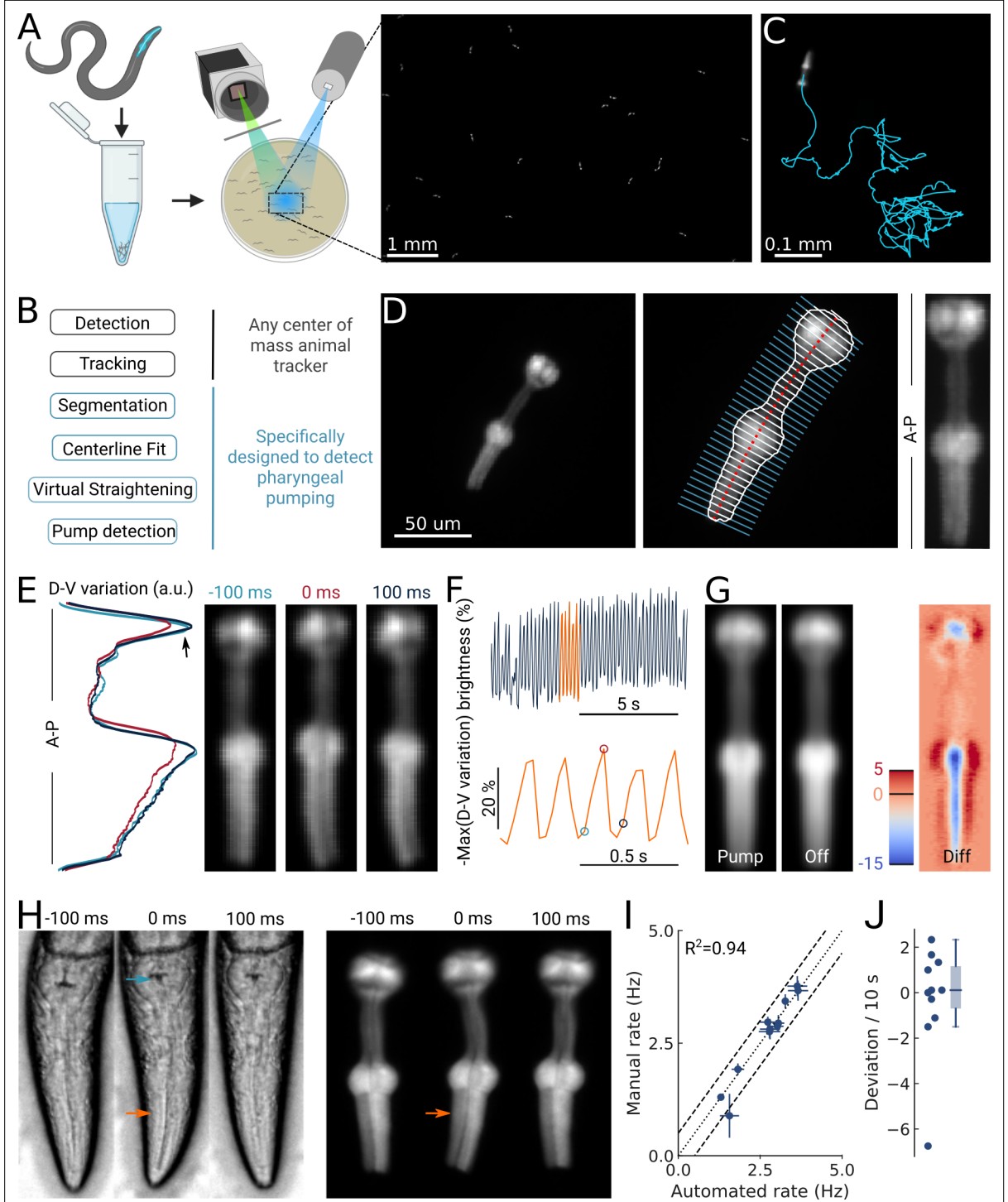

**Figure 1.** High-throughput optical detection of pharyngeal pumping in moving worms. (**A**) Hundreds of animals expressing *myo-2p::YFP* are washed in M9 and pipetted onto the assay plate before imaging with an epi-fluorescence microscope at x1 magnification resulting in a full field of view of 7 by 5 mm. (**B**) Workflow of using the PharaGlow image analysis pipeline. Animal center of mass tracking can be substituted with any available tracker, but subsequent steps are specific to tracking pumping. (**C**) Representative trajectory of an animal after tracking. (**D**) Processing steps followed for detection of pharyngeal pumping. Example of a fluorescent image (left; 2x magnification). Segmentation of pharyngeal contour, centerline, and widths (middle) calculated for virtual straightening along the anterior-posterior axis (**A–P**) and the resulting straightened animal (right). (**E**) Three straightened frames of an animal before, during, and after a pump and their dorso-ventral variation in brightness along the A-P axis. (**F**) The metric that is used to detect pumping events. Bottom, a portion of the top trace (orange). Highlighted time points correspond to the images in (**E**). (**G**) Average of all images during a detected pump ('Pump') and for all remaining timepoints ('Off'). The difference image ('Diff') shows that pumps are characterized by the opening

*Figure 1 continued on next page*

*Figure 1 continued*

of the lumen and terminal bulb contraction. Colorbar indicates brightness difference (a.u.). (**H**) Example image sequence of a pharynx recorded at 10 x using bright-field (left) and in epi-fluorescence (right) microscopy before, during, and after a pump. Arrows denote changes in the terminal bulb (cyan) and corpus (orange). (**I**) Correlation between the average pumping rates for the expert annotator and PharaGlow (N=11 animals). (**J**) Deviation of the number of events between the expert and PharaGlow reported as the number of events in 10 s, a typical time period used in manually counted experiments.

to adulthood (*Avery and Shtonda, 2003*). To investigate how pumping rates change during development, we imaged cohorts of synchronized worms consecutively over three days in the middle of each of the four larval stages and as young adults (YA). Animals were imaged directly on their culturing plates while moving freely in the field of view (*Figure 2A*). We accounted for the growing pharynx by adapting the magnification of our imaging system to achieve approximately the same spatial sampling of the pharynx at each stage (*Figure 2B*). Under these conditions, we were able to sample at least 150 trajectories per developmental stage. Altogether, more than 1000 animal tracks remained after filtering animals that spend less than one minute in the field of view. Filtering leads to overproportionally reducing young adult trajectories since these animals traverse the field of view quickly despite the spatially proportional scaling. Nevertheless, we obtain large samples of animals due to new animals continually entering, with a total measured time of four animal-hours for the adult stage, and more than 10 animal-hours for the L1 stage. The average track duration is well over one minute with 1.9 ± 0.9 min (mean ±s.d.) for L1 and 1.6 ± 0.6 min for adults (*Figure 2—figure supplement 1*). These data represent up to two orders of magnitude more single worm pumping data than is obtainable with conventional methods (see *Table 1*).

We find that on-food pumping rates increase slightly over the course of the larval stages, but much less dramatically than the velocity increases over development, despite the substantial growth of both the body and the pharyngeal muscles (*Figure 2C–F*). Owing to time resolution and the large number of individual worms that can be analyzed using PharaGlow, it is possible to generate smooth probability density functions of pumping across the different larval stages (*Figure 2G*). A small fraction of animals did not show pumping during our recording (*Figure 2H*, 5 animal tracks in L1 with <0.5 Hz, <1% for all other conditions). We wondered if we had captured animals during lethargus, the period of sleep preceding each molt despite choosing the imaging time points in the middle of each larval stage and working with an age synchronized population. However, lethargus is incompatible with the observed velocities of these animals. Alternatively, it is possible that these animals transiently show satiety quiescence, which might be absent under these conditions in the larger YA population (*You et al., 2008*; *Gallagher et al., 2013*; *Davis et al., 2018*).

As we image unrestrained animals, we can simultaneously assess pumping and locomotor behaviors. Animals move forward on agar by generating waves of muscular contraction through their body. When the animals reverse the direction of these waves, they move backwards. Such spontaneous reversals are rare events, but can be triggered by diverse stimuli, such as nose touch (*Chalfie et al., 1985*) or heat (*Zhao et al., 2003*). The reversal rate depends also on the food condition and the developmental stage of the animal. In the absence of food, the reversal rate is higher in young adults than in larvae (about 45 events vs 30 events in 10 min), but constant throughout

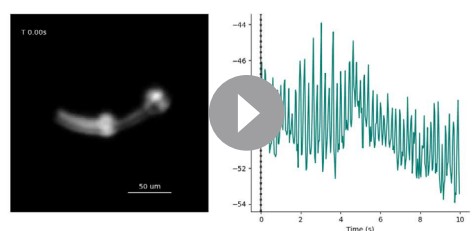

**Video 1.** GRU101 worm pumping with the pumping metric shown.

https://elifesciences.org/articles/77252/figures#video1

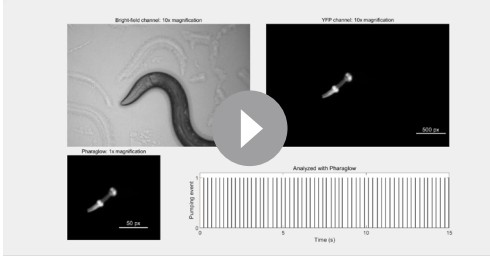

**Video 2.** Simultaneous bright-field (top-left) and fluorescence imaging (top-right) of a freely moving GRU101 worm at ×10 magnification. Downscaled fluorescent images (×1 magnification; bottom-left) and resulting pumping events detected by PharaGlow (bottom-right).

https://elifesciences.org/articles/77252/figures#video2

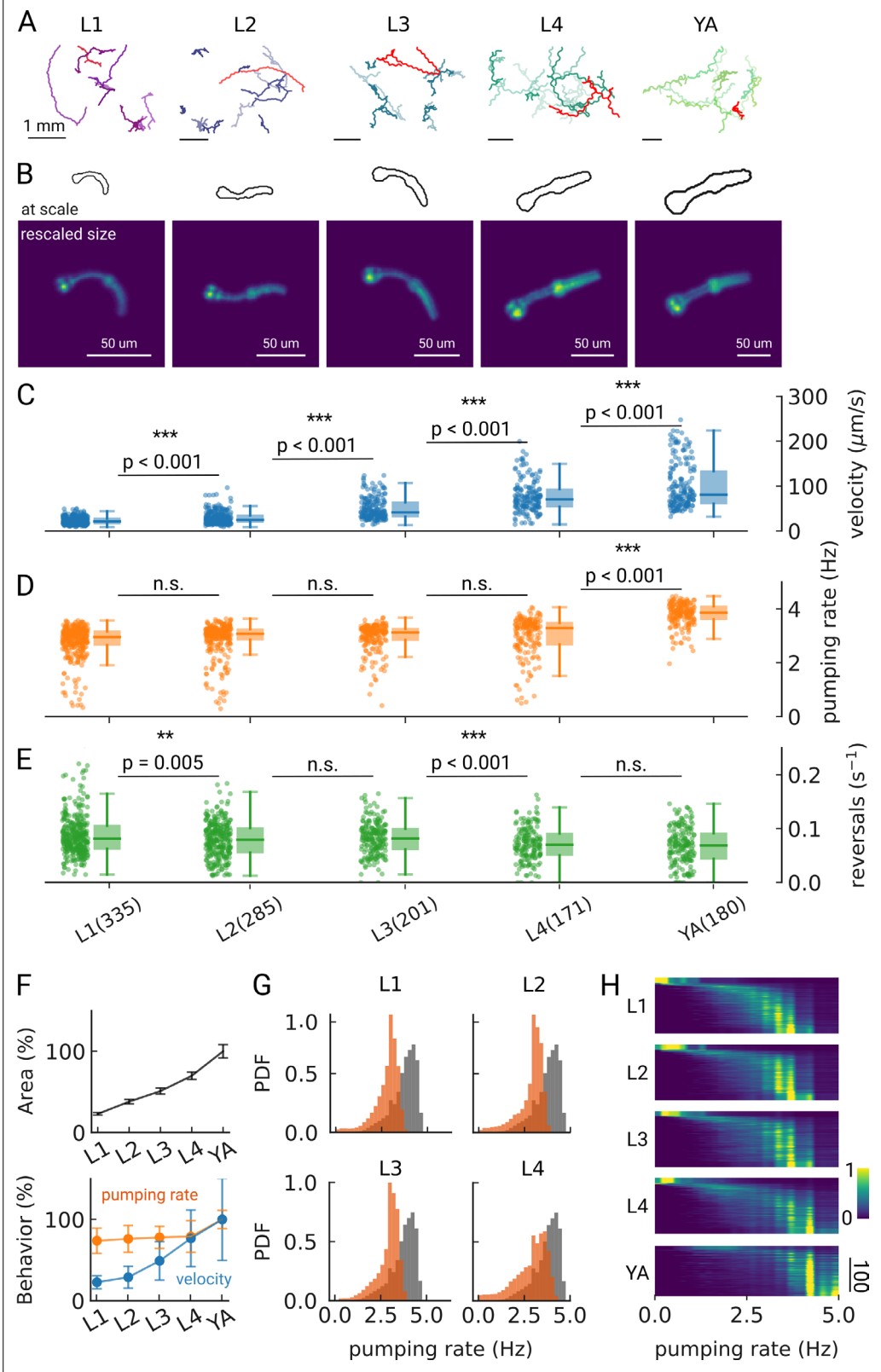

**Figure 2.** Changes in pumping and locomotion during larval development. (**A**) Trajectories of 10 randomly selected animals at different larval stages (**L1–L4**) and young adults (YA). All scale bars correspond to 1 mm (top). (**B**) Size of the larvae and YA at the same scale (outlines, top) compared to the equal sizing achieved by adjusting the magnification (bottom). The image corresponds to the red track from (**A**). (**C–E**) Time-averaged mean velocity

*Figure 2 continued on next page*

*Figure 2 continued*

(**C**), (**D**) mean pumping rate, and reversals (**E**) for all animals. The boxplots follow Tukey's rule where the middle line indicates the median, the box denotes the first and third quartiles, and the whiskers show the 1.5 interquartile range above and below the box. The number of tracklets per developmental stage are shown in (**E**), with N=6 independent replicates per condition. (**F**) Relative change in the animal's area compared to the mean area of the YA stage (top) and relative change in velocity (blue) and pumping rate (orange) across development compared to the mean of the YA stage (bottom). Error bars denote s.d. (**G**) Pumping rate distribution for all larval stages as calculated by counting pumping events in a sliding window of width = 10 s and combining data from all animals of the same stage. The YA pumping rate distribution is underlaid in gray. (**H**) Pumping frequency distribution of individual worms for different developmental stages and YA.

The online version of this article includes the following figure supplement(s) for figure 2:

**Figure supplement 1.** Track duration and number of tracked animals per experiment.

**Figure supplement 2.** Reversal rates across development.

**Figure supplement 3.** Only a mild leaving was induced by light.

**Figure supplement 4.** Possible light-induced behavioral changes.

**Figure supplement 5.** Pumping detection is robust at two different excitation wavelengths.

**Figure supplement 6.** Detection accuracy for all developmental stages.

---

larval development (*Chiba and Rankin, 1990*). In our on-food measurements, we find some significant differences in reversal rates, however, the effect size is small (e.g. corresponding to a rate of 47 vs 49 pumps/10 min between L2 and L3 animals). The only strong difference appears between the earlier larvae L1-L3 and the later L4/young adult stages with a difference of approximately 10 reversals /10 min (*Figure 2—figure supplement 2*).

We investigated whether extended exposure to light might affect worm behaviors by monitoring the amount of reversals, pumping rate, and speed. Our results suggest that there is a mild light avoidance reaction (5–25%) which depends on the developmental stage (*Figure 2—figure supplement 3*). Repeated exposure did not affect behavior in most developmental stages, except for the young adult stage, for which a higher velocity was observed when exposed to light multiple times (*Figure 2—figure supplement 4*). Additionally, we tested whether light exposure caused phototoxic effects. Long-term exposure (up to 5 hr) did not affect worm viability. Lastly, different excitation light did not affect the pumping rate, but had a mild impact on velocity (*Figure 2—figure supplement 5*; see Appendix for details).

Overall, we find that our imaging approach can be adapted to larvae by increasing the magnification, and our analysis pipeline is capable of handling data from hundreds of animals. While there are small deviations between the automated detection and human counted data (*Figure 2—figure supplement 6*), we accurately detect both mean and individual rates for all stages, with a median of error between experts and our method of less than 10%. Over the course of three days and five imaging sessions, more than 1000 animals were tracked, significantly more than can be achieved with comparable methods (*Table 1*).

## Food intake is modulated by starvation

Next, we wanted to determine if our method was robust to changes in locomotion and plate context, allowing a wider range of applications such as investigating starvation or different pharmacological treatments without the presence of a bacterial food source. Off-food locomotion is faster (*Dillon et al., 2016*; *Gray et al., 2005*), and pumping irregular (*Lee et al., 2017*; *Scholz et al., 2016*), which could potentially be more challenging for detecting pumping. Prior work showed that pumping rates off food are lower, but increase over the course of starvation and that this increase is mediated by a cholinergic pathway (*You et al., 2006*). We track animals either on-, or off food over an increasing amount of starvation time and extract behavioral dynamics (*Figure 3A–D*). We confirm that pumping is dependent on the starvation duration, with a reduction in pumping rate over the course of three hours (*Figure 3B, D*). Beyond the first time point, our data are consistent with prior data (*You et al., 2006*), showing a sustained rate of around 2–2.5 Hz (*Figure 3E*). Previously, rates measured immediately after transferring worms off food (<30 min of starvation) were very low, possibly due to a lasting

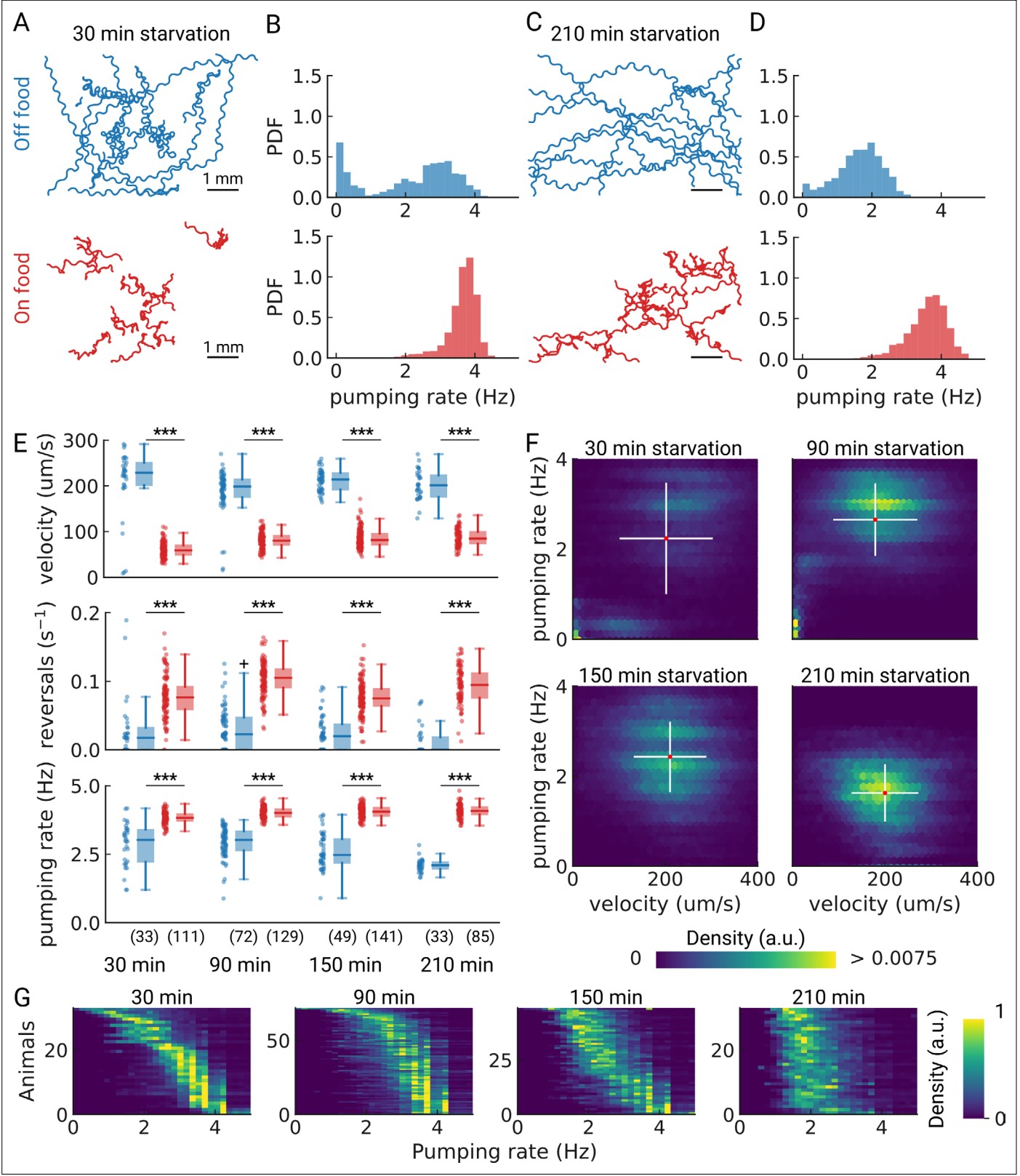

**Figure 3.** Pumping is modulated by starvation. (**A**) Example trajectories of worms after 30 min starvation (blue) or 30 min continuously on food (red), N=10. (**B**) The pumping rate distributions for the conditions in (**A**) for all animals ($N_{starved}$ = 33, $N_{onFood}$ = 111). (**C**) Same as (**A**) but for animals starved, or kept on food for 210 min. (**D**) The pumping rate distributions corresponding to (C; $N_{starved}$ = 33, $N_{onFood}$ = 85). (**E**) Velocity, reversal rate, and pumping rate for animals starved and on-food controls. The sample size is given in the bottom panel. *** indicates $P<0.001$ (Welch's unequal variance two-tailed t-

*Figure 3 continued on next page*

*Figure 3 continued*

test). The sample size is given in parentheses in the bottom panel. (**F**) Joint distribution of velocity and pumping rate for increasing starvation times. The cross indicates the mean (red) and standard deviation (white). The density is normalized by sample number. (**G**) Distribution of instantaneous pumping rates for each animal (tracklet). Rows are sorted by the mean pumping rate to aid visualization.

The online version of this article includes the following figure supplement(s) for figure 3:

**Figure supplement 1.** Pumping and velocity correlation on food.

pumping suppression after harsh touch (*Keane and Avery, 2003*), which we avoid by washing worms off plates instead of picking (see Methods).

As we are able to measure pumping and locomotion behaviors simultaneously, we wanted to see if we could observe co-regulation of locomotion and feeding off-food. When taken off of food, *C. elegans* displays a restricted area search (local area search) which is characterized by frequent turns and reversals and an elevated speed (*Gray et al., 2005*; *Hills et al., 2004*; *Sawin et al., 2000*; *Calhoun et al., 2014*). This behavior lasts between 30 and 60 min, after which animals switch to longer runs that cover more area, which is a strategy for dispersal (*Hills et al., 2004*; *Wakabayashi et al., 2004*). Interestingly, for starved worms at 30 and 90 min, the joint distribution of pumping rates and velocities show distinct sub-populations (*Figure 3F*). For longer starvation durations, the population becomes homogeneous with a well-defined mean pumping rate and speed. For the shortest starvation time point we sampled, we see a mixed population with distinct speeds and pumping rates, possibly reflecting some animals that are still performing a local search and others that are not. This is consistent with the fact that these distinct populations are not apparent in worm populations that stay on food (*Figure 3—figure supplement 1*).

To further investigate the origin of the two sub-populations observed at short starvation time, we analyzed the pumping rate distributions of individual animals (*Figure 3G*). Taken together, the data suggests that at 30 min starvation, a fraction of the animals show low speeds and pumping rates, and the remainder are in a high-speed, high pumping state (*Figure 3F, G*). This suggests two possible interpretations. First, it is possible that, with increasing starvation time, a subset of animals transitions to lower pumping rates until all animals show a similar pumping rate distribution with an average of ~2 Hz. Alternatively, the two sub-populations could result from transient behavioral changes among animals to high pumping rates. These transitions would occur less frequently with increasing starvation time. To discern among these two possibilities would require measuring single animals over longer periods of time. Further studies are required to reveal these population dynamics upon starvation.

## Long-term recording of mating animals

Having established that PharaGlow can robustly detect locomotion and pumping behaviors across a range of conditions, we wanted to test if it is a suitable tool for long-term recordings. As a proof of principle, we imaged the interactions of a male and a hermaphrodite over the course of 74 min at 30 fps (*Figure 4A and B*).

As the resulting data volume would have been prohibitive, we implemented a live segmentation method that allowed us to only store the animals coordinates and the region of interest around each animal (*Videos 3 and 4*; Methods). We then calculated the distance between the animals, allowing us to identify mating events (*Figure 4C and D*). We find that the animals frequently interact over the course of 1 h with multiple close encounters (*Figure 4D*). The male also showed a long period of quiescence in both locomotion and pumping rate. Overall, the animals are closer at the beginning of the recording, but later spend time at larger distances (*Figure 4E*, left and right panels). Despite the long imaging duration, we still observe pumping at the end of the recording, indicating that we have sufficient signal remaining to detect pumping events. We confirm this observation by calculating photo-bleaching curves. We find that the decay time of the signal is 410±47 min (*Figure 4—figure supplement 1*), which indicates that it is possible to do continuous imaging over multiple hours. In this case, the recording was limited by the male escaping the enclosure, rather than loss of signal.

Having this multi-scale data allows observing both large-scale structure and smaller events in the data. We further examined the mating event displayed in *Figure 4C*. During the encounter, the male shows a larger velocity and performs many long reversals when the animals are close, as is typical for a mating attempt. It is also interesting to note that pumping does not completely cease during the attempt (*Figure 4F*). Despite being in the same arena, and covering most of the enclosure

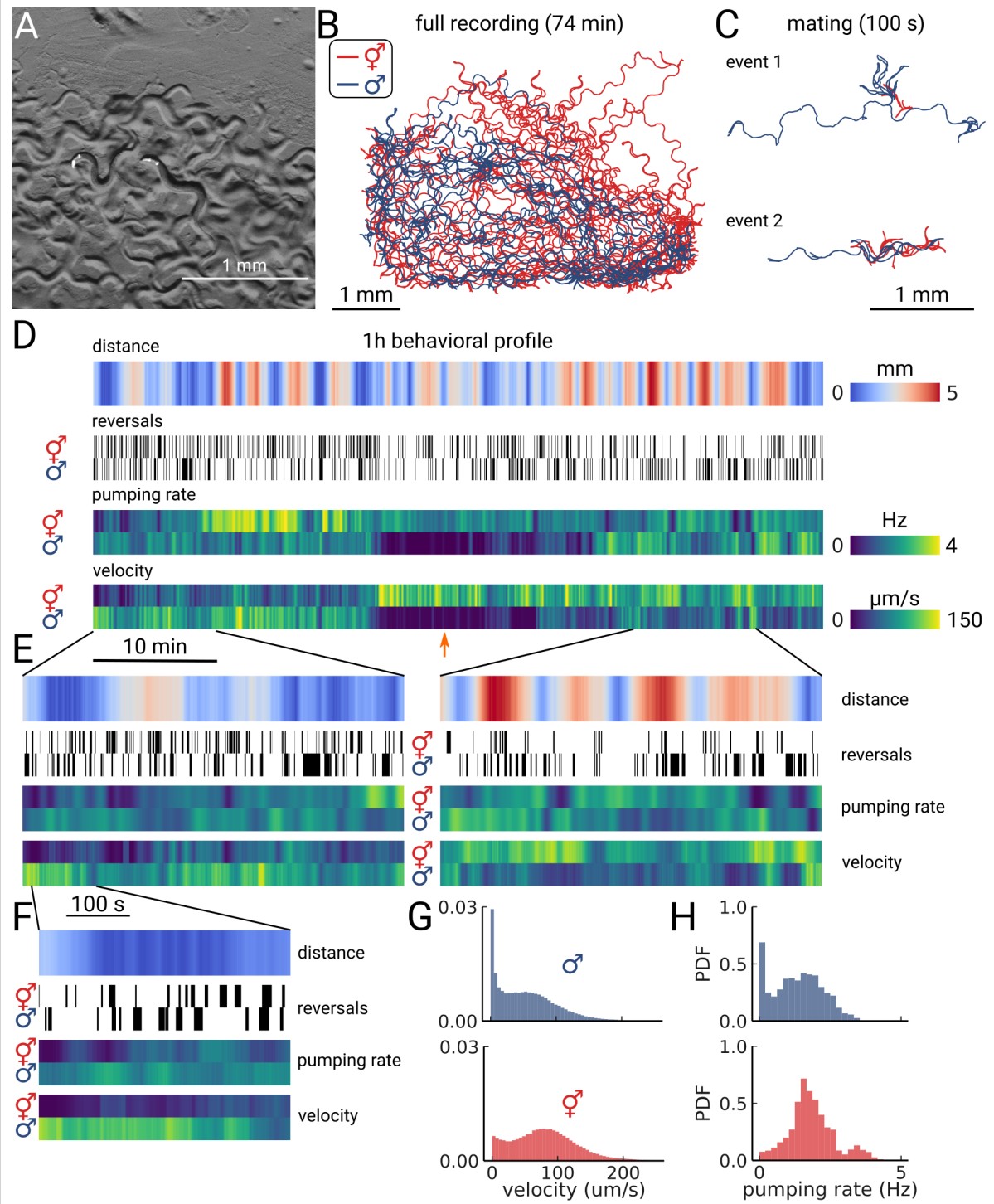

**Figure 4.** Long-term imaging of mating animals. (**A**) Composite image of the two animals in the arena while exposed to bright-field illumination and exciting fluorescence of YFP using green light. On the right, the hermaphrodite, on the left the male identifiable by its smaller size and its tail with sensory rays and fan. (**B**) Trajectories obtained from the full recording of the male (blue) and hermaphrodite (red). (**C**) Example mating events. (**D**) Behavioral measures for 1 hr of data. The distance between the animals, the reversal events, pumping rate, and velocity are shown for the hermaphrodite and the male. The male shows an extended period of quiescence (orange arrow). (**E**) Behavioral measures for 10 min of data and (**F**) 100 s of data corresponding to the mating event 1 in panel (**C**). (**G**) Velocity distribution and (**H**) pumping rate distribution for the male and hermaphrodite.

The online version of this article includes the following figure supplement(s) for figure 4:

**Figure supplement 1.** Bleaching of YFP.

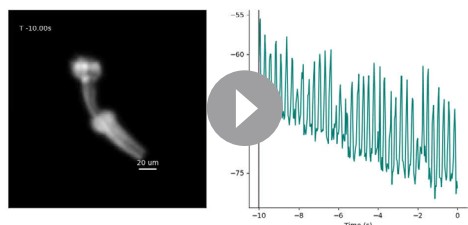

**Video 3.** *eat-18* worm pumping.
https://elifesciences.org/articles/77252/figures#video3

during the recording (*Figure 4B*), the velocity and pumping distributions differ strongly between the two animals (*Figure 4G and H*). While the distributions of the male are dominated by the long quiescence period, the hermaphrodite overall shows a bi-modal rate distribution with some infrequent pumping at 4 Hz. PharaGlow is therefore able to track behavior over more than an hour, and keep the identity of animals given that these are constrained to the field of view.

## Feeding mutants

A desired capability for a high-throughput feeding tool is the ability to faithfully detect pumping rates in mutant animals which might have different pharyngeal contraction patterns and body motion, potentially increasing the difficulty of detecting pumping events. To determine if PharaGlow could faithfully detect pumping and locomotion in mutant animals, we wanted to assay a range of feeding and locomotion phenotypes. We therefore selected mutants with reported constitutively high (*unc-31*) or reduced (*eat-18*) pumping rates and different locomotory patterns (*Raizen et al., 1995*; *McKay et al., 2004*; *Avery et al., 1993*). UNC-31 is involved in dense-core vesicle release, and *unc-31* mutant animals display reduced, uncoordinated locomotion on food (*Figure 5A*). We confirm that *unc-31(e928)* and *unc-31(n1304)* animals pump at rates comparable to wildtype. However, we see a bimodal distribution of rates with a fraction of animals showing markedly lower rates (*Figure 5B*). By looking at the individual animals' pumping rates, we find that *unc-31* animals show long pauses in pumping, unlike wt animals (*Figure 5F*).

In contrast to *unc-31*, *eat-18* mutant animals have no previously reported locomotor defects, but pump slower than wildtype (*McKay et al., 2004*). EAT-18 is expressed in the pharyngeal muscle and interacts with a nAChR subunit EAT-2 to form a functional acetylcholine receptor (*Raizen et al., 1995*; *Choudhary et al., 2020*). Feeding impaired mutants were previously reported to have reduced body lengths and widths (*Mörck and Pilon, 2006*). As expected, we found that *eat-18* animals were smaller (*Figure 5C*) and developed more slowly (approximately 91 hr from egg to adulthood compared to 63 hr for wildtype). While we detected pumping events at an average rate of 1 Hz, the animals showed a different contraction pattern and timing than either *unc-31* or wt animals (*Figure 5D and E*). We confirmed that *eat-18(ad820)* animals lack the ability to perform fast pumping bursts (*Figure 5B and F*) and the duration of a pharyngeal contraction is approximately doubled compared to wt (*Figure 5E*, *Figure 5—figure supplement 1*, *Video 3*). We do observe a higher pumping rate than previously reported for *eat-18*, where animals were reported to rarely pump during experiments (<0.5 Hz, *Raizen et al., 1995*; *McKay et al., 2004*). To verify that the detected motion is pumping and not peristaltic movements or other non-pharyngeal muscular motion, we verified the rate by inspecting individual videos. When verifying these sample animals, we did observe slow pumping bursts at the 1–2 Hz rates indicated by PharaGlow, suggesting that these animals are able to pump at this frequency (*Figure 5E*, arrows and *Video 3*). We also found that *eat-18* animals showed significantly fewer reversals than wildtype, indicating a role for the nAch receptor in modulating reversals. This is likely mediated by extrapharyngeally located neurons, since *eat-18* is reported to show expression not only in the pharyngeal muscle, but also in some unidentified somatic neurons (*McKay et al., 2004*).

Considering the split distribution of mean pumping rates we observed for *unc-31* in our short term (5 min) recordings (*Figure 5B*), we wondered if these distributions reflect a persistent difference between animals or if instead the animals perform

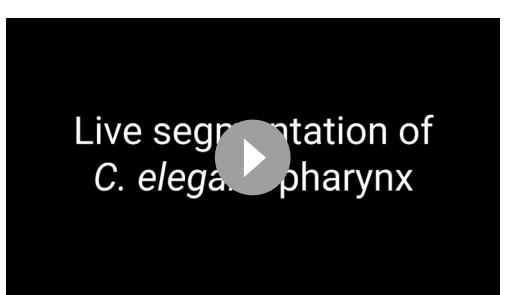

**Video 4.** Live segmentation of worms. The program stores only the segmented worms from individual images and their xyzt coordinates to reduce storage requirements by several orders of magnitude thus allowing uninterrupted recordings for hours at 30 FPS.
https://elifesciences.org/articles/77252/figures#video4

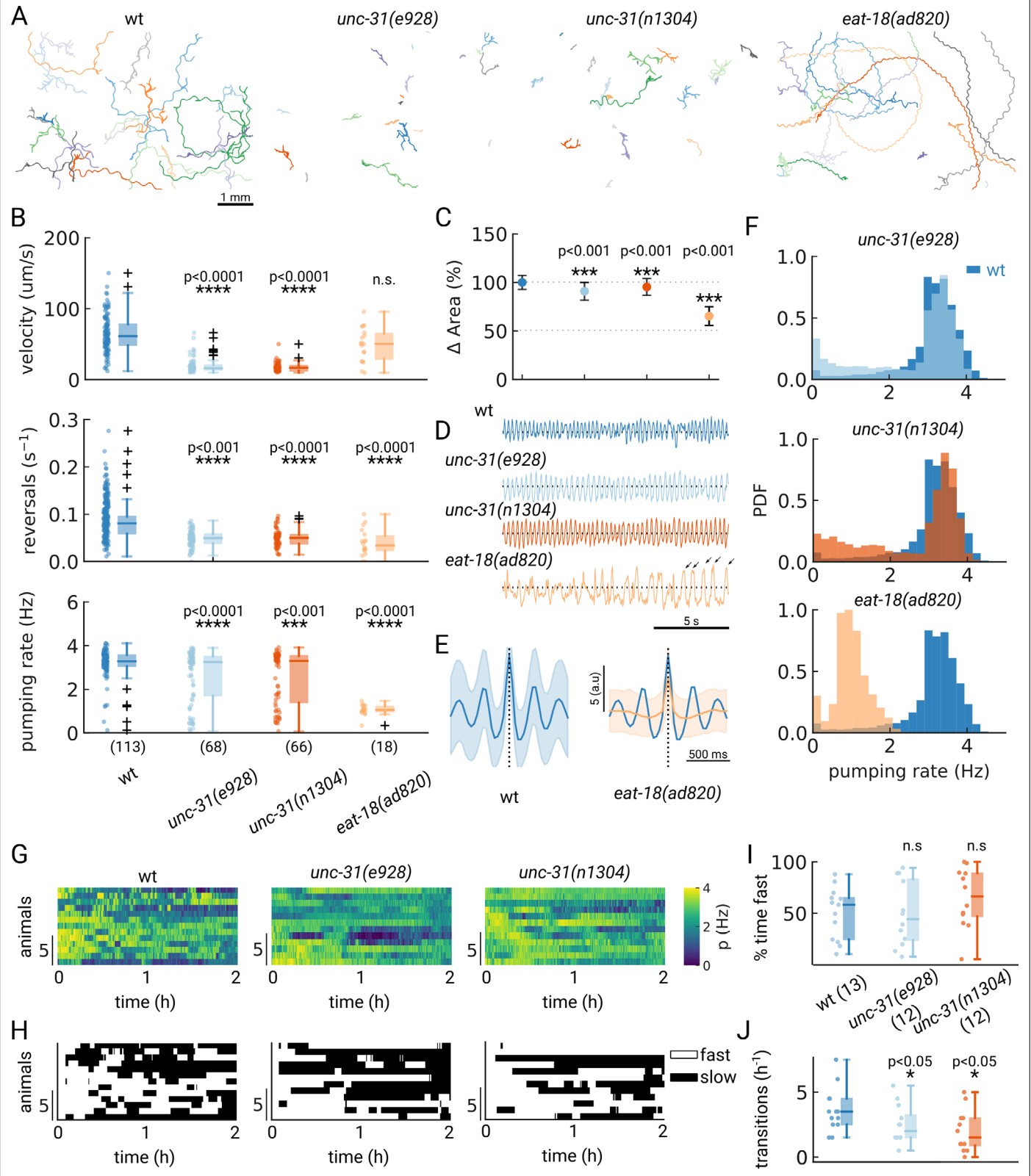

**Figure 5.** Automated pumping detection in feeding mutants. (**A**) Example trajectories of tracked animals (N=20, except N=18 for *eat-18(ad820)*). (**B**) Velocity, reversal rate and pumping rate for all genotypes. The sample size is given in parentheses in the bottom panel. (**C**) Mean and standard deviation of the pharyngeal areas relative to wt. (**D**) Pumping metric for a representative sample animal per genotype. Arrows in the *eat-18(ad820)* trace denote slow contractions. (**E**) Average peak shape of the pumping signal for wt (blue) and *eat-18(ad820)* (orange). The shaded area denotes s.d.

*Figure 5 continued on next page*

*Figure 5 continued*

(**F**) Pumping rate distributions. The wt pumping rate distribution is underlaid in dark blue. (**G**) Heatmap of the pumping rates for animals recorded over 2 h. (**H**) Heatmap thresholded to determine 'fast pumping' (defined as pumping rate >2.5 Hz) and 'slow' states. (**I**) Fraction of time spent in fast pumping of each animal and (**J**) the number of state transitions (slow to fast and fast to slow) for each animal in (**H**). Significant differences between a mutant and wt are indicated as * (p<0.05), *** (p<0.001) and **** (p<0.0001). Welch's unequal variance two-tailed t-test was for the large sample size measurements (**B**, **C**). For (**I–J**) significance differences were assessed with the Mann-Whitney-U test.

The online version of this article includes the following figure supplement(s) for figure 5:

**Figure supplement 1.** Peak-triggered average of the pumping metric.

**Figure supplement 2.** Pumping rate autocorrelation.

infrequent switches between high - and low pumping rate states. We therefore tracked animals for at least three hours on food, restrained to our field of view using a copper enclosure. By confining only a few animals (<5 in the field of view), we were able to maintain animal identity over the course of the experiment (see Methods) and quantify their pumping rate over at least 2 hr (*Figure 5G*). If the population reflects a snapshot of the overall dynamics, a large population measured for a short period of time should result in similar pumping rates as a few animals measured over long time periods. To test this hypothesis, we quantified the autocorrelation of the pumping rate for single worms revealing that pumping rates were correlated for different time scales for *wt* and *unc-31* mutants, with the *unc-31* mutants showing a more persistent pumping behavior ($\tau$=9.6 min for *unc-31(e928)* and $\tau$=11.0 min for *unc-31(n1304)*) compared to wt ($\tau$=7.9 min) (*Figure 5—figure supplement 2*). These data indicate that on-food pumping rates are autocorrelated over multi-minute time-scales.

To further investigate the different persistence of pumping across the different animals, we quantified the transitions between states of pumping and no or low pumping rates. We define a 'fast' state as a period at which the animals pump at >2.5 Hz and 'off' states as the converse (see also a similar analysis in *Lee et al., 2017*). Wild-type animals displayed frequent switching between low and high pumping rates (*Figure 5H and J*). In contrast, *unc-31* animals displayed infrequent switches, consistent with prior reports of constitutive pumping (*Avery et al., 1993*) and the role of neuropeptides such as PDF in regulating switches between foraging states (*Flavell et al., 2013*). Taken together, these results show that studying the underlying behaviors and dynamics in a worm population requires large statistics and long recordings. Depending on the desired data, both long-term recordings and short-term high-throughput measurements are accessible with PharaGlow.

## Limitations and requirements

Using a combination of low-magnification fluorescence imaging and dedicated analysis software, we show that it is possible to perform high-throughput, automated pumping detection of worms crawling on standard culture plates. There are some limitations to the method that are due to the reliance on a fluorescent indicator in the pharyngeal muscle and the handling of the large datasets that are generated. We find that uneven plates or improper focus leads to low signal-to-noise ratios. With careful focusing, imaging the center of evenly poured plates and using our custom peak detection method that is adaptive to the image quality (see Methods), these pitfalls can be mitigated. Additionally, once at focus, small variations of the worm height do not affect the result, as low-magnification imaging results in a large depth of field.

As it is necessary for our approach to label the pharynx, mutant characterization with our tool would require crossing all possible mutants with a fluorescent reporter strain, and albeit labor intensive this is nonetheless still a standard genetic practice when a reporter is used. While most experiments in this paper were performed with a homozygous, integrated reporter background (*gnaIs1*), we have also used extrachromosomal arrays with success (*Figure 2—figure supplement 5*), which allows the use of animals that have a *myo-2* reporter as a co-injection marker, for example. In addition, since our tool relies on the detection of fluorescent protein, siblings losing the transgene on the plate will not interfere with the analysis.

To maximize the field of view, we have chosen the smallest spatial resolution at which we could reliably detect pumping in wildtype adults. To ensure detection in smaller animals, increasing the magnification in these cases is recommended, as we did to detect pumping in larvae. The final requirement is related to data management and handling. While the hardware requirements are restricted to equipment commonly available in many laboratories (a fluorescence dissecting microscope or a

epifluorescence microscope and a Megapixel camera are needed), the data rate of the large area scan cameras is >6 GB/min. On a 4-core laptop, the expected analysis time is approximately 8 hr for 150 worm-minutes of data. While the analysis can be run on a laptop or desktop computer, runtime is much improved when using a computing cluster (see Supplementary materials).

## Discussion

We developed a microscopy protocol and customized image analysis software that enables simultaneous measurements of locomotion and feeding in unrestrained animals. We are able to accurately detect pumping rates in populations of animals on plates at single-worm, single-pump resolution. Our fully automatic method provides an increase in throughput by more than one order of magnitude, and does not require any laborious handling of the animals or microfluidic devices. As our imaging does not adversely affect worms, animals can be imaged multiple times, as we demonstrated by following pumping in a population of developing animals or can be imaged continuously for hours as shown for the mating animals. In addition, by enabling measurements of unrestrained animals directly on agarose plates, our approach creates opportunities for studying novel behaviors, such as foraging in complex environments (*Ding et al., 2020*; *Iwanir et al., 2016*) or behavioral coordination (*Cermak et al., 2020*; *Hardaker et al., 2001*).

Using our method, we were able to detect quiescent episodes in all four larval stages that were absent in the young adult. It will be interesting to study the food needs at different stages and how feeding is regulated on the scale of minutes. To understand the neural basis of feeding, resolving the decisions underlying feeding behavior is required (*Scholz et al., 2017*; *Katzen et al., 1983*). However, current bulk methods capable of extracting population averages of pumping rates are insufficient to understand underlying neural activity, as the data lack the temporal resolution to correlate pumping with neural activity. Additionally, it is desirable to image animals in their normal culturing environments to compare feeding behaviors to established baselines for example, for velocity and reversals. Our method enables the quantification of pumping activity, in multiple animals simultaneously, as a marker of feeding behavior in *C. elegans*.

An essential requirement for the broad applicability of such a tool is its use for genetic and pharmacological screens. We demonstrated the suitability of our tool for studying the pumping and locomotory behaviors of *unc-31* and *eat-18* animals. We could identify a bimodal distribution of pumping rates for *unc-31* mutants with some animals showing low rates for the duration of the recording. This indicates that while these animals are capable of fast pumping, they do not show the same temporal regulation as N2 animals, which show rapid transitions between slow and fast pumping, and pump fast most of the time at high food levels (*Scholz et al., 2017*). We could also reveal a previously unreported alteration in pump duration for *eat-18* animals. Surprisingly, during confirmation of the feeding defect, we also discovered a previously unreported locomotion defect, hinting at a broader function for *eat-18* possibly outside of the pharynx.

Additionally, PharaGlow enables easy pharmacological screens on plates, by allowing experimenters to directly apply compounds to culture plates without requiring microfluidics or immobilization. Immobilization often requires the addition of serotonin (5-HT) to stimulate pumping, as pumping is suppressed in restrained animals (*Takahashi and Takagi, 2017*). It is likely that observed feeding defects differ between crawling, food-stimulated animals and serotonin-stimulated animals (*Lee et al., 2017*), and that new experiments will be effective in identifying phenotypes that went unnoticed in immobilized preparations.

To mimic realistic foraging situations would require extending the foraging arena by supplying variable patches of food, and providing a more interesting landscape rather than a homogenous 2D plate environment. Patch foraging has been extensively studied in *C. elegans*, both with respect to entry into single patches (*Katzen et al., 1983*; *Flavell et al., 2013*; *Iwanir et al., 2016*) and food choice (*Katzen et al., 1983*; *Milward et al., 2011*; *Dal Bello et al., 2021*), although these studies focused on locomotion rather than pumping rates. Testing models of patch foraging and their predictions (*Davidson and El Hady, 2019*) while explicitly including measured food intake will help better define the limits in which simple models of foraging are applicable. Moreover, this would ideally expand our understanding of the underlying strategies learned or inherited by worms living in different environments. Technically, current imaging limitations, such as the field of view, can be extended by using larger camera arrays, which will enable scanning across multiple centimeter-sized fields of view. These

experiments would also require long-term imaging of the animals to observe transitions between food patches, which is possible using our method.

A further extension of this work would be to image pumping activity in related species, either *Caenorhabditis* nematodes collected from different field sites, or even predatory nematodes like *Pristionchus pacificus,* which has a bacterial feeding mode similar to *C. elegans,* and additionally a predatory mode when killing the larvae of other nematodes. While the requirement for labeling the muscle is a prerequisite of our method, the *myo-2* gene is a myosin heavy chain that is conserved among nematodes and will likely show similar expression in closely related species. A strain with the *myo-2* promoter construct for the closely related *C. briggsae* is already available at the CGC. Our tool opens new venues to study feeding behaviors at multiple scales. We envision that its application will lead to new insights into worm behavior.

# Materials and methods

**Key resources table**

| Reagent type (species) or resource | Designation | Source or reference | Identifiers | Additional information |
|---|---|---|---|---|
| Strain, strain background (*Escherichia coli* OP50) | OP50 | CGC | CGC:OP50 | |
| Recombinant DNA reagent | *myo-2p mCherry unc-54 3'utr* | Addgene | pCFJ90 | 5 ng/µl injected into N2 |
| Recombinant DNA reagent | *pPHA2 GFP-F* | Gift from Marc Pilon | pMS17 | 50 ng/µl injected into N2 |
| Strain, strain background (*C. elegans*) | N2 | CGC | N2 | Background for INF30 |
| Strain, strain background (*C. elegans*) | *gnaIs1[myo-2p::yfp]* | CGC | GRU101 | |
| Strain, strain background (*C. elegans*) | *nonEx9[pPHA2 GFP-F myo-2p::mCherry::unc-54 3'utr]* | This publication | INF30 | 5 ng/ul of pMS17 and 5 ng/ul of pCFJ90 injected into N2; *Figure 2—figure supplement 4* |
| Strain, strain background (*C. elegans*) | *unc-31(e928) gnaIs1 IV* | This publication | INF5 | Cross of *unc-31(e928)* with GRU101; *Figure 3* |
| Strain, strain background (*C. elegans*) | *unc-31(n1304) gnaIs1 IV* | This publication | INF17 | Cross of *unc-31(n1304)* with GRU101; *Figure 3* |
| Strain, strain background (*C. elegans*) | *eat-18(ad820) I; gnaIs1 IV* | This publication | INF44 | Cross of *eat-18(ad820)* with GRU101; *Figure 3* |
| Software, algorithm | PharaGlow | This publication | | https://github.com/scholz-lab/PharaGlow; *Scholz, 2022*. |

## *C. elegans* maintenance

*C. elegans* were grown on NGM plates at 20 °C. Worms were synchronized by letting adult gravid animals lay eggs for 2–3 hr, then removing the adults. The average time from egg to young adult stage for strain GRU101 (*gnaIs1[myo-2p::yfp]*) was 63 hr. Before the experiment, synchronized adults were washed off the culture plates with 1 ml of M9 and collected in an Eppendorf tube. Worms were allowed to settle for 1 min, the supernatant was removed and the tube was refilled with M9. Washing was repeated two more times. The washing was sufficient in that we did not observe animals remaining in the spots containing the remainder of M9 on the assay plates, suggesting that the bacterial amount was too diluted to induce dwelling behavior.

## Imaging setup

Imaging of worms at ×1 magnification was performed using a commercial upright epi-fluorescence microscope (Axio Zoom V16; Zeiss) equipped with a 1 x objective (PlanNeoFluar Z 1.0 x/N.A. 0.25). For imaging of YFP fluorescence, light from an LED lamp (X-Cite XYLIS) was reflected towards the sample using a dichroic mirror (FT 515; Zeiss) and filtered (BP 500/25; Zeiss). Emitted light was filtered using a band-pass filter (BP 535/30; Zeiss) and focused onto the camera sensor (acA3088-57um; BASLER) using a camera adapter with an additional 0.5 x magnification (60 N-C ⅔" 0.5 x; Zeiss). The power

density of fluorescence excitation at the focal plane (0.24 mW/mm² at 500 nm) was measured using a power meter sensor (PS19Q; Coherent) with the corresponding controller (PowerMax; Coherent). Animals were imaged at 30 fps for 5 min unless otherwise indicated. For imaging of mCherry, the filter cube was replaced with a commercial filter set (64 HE; Zeiss). The resulting power density using this cube was 0.76 mW/mm² at the focal plane.

### Long-term imaging

For long-term imaging, stroboscopic illumination (5ms light pulses) were used to reduce bleaching. Excitation light was synchronized with the camera exposure using the GPIO camera line and the TTL input of the LED lamp. Frames were collected using a custom software (LabVIEW). To reduce the amount of stored data and allow continuous recording using a standard computer (Celsius W520; Fujitsu), images were segmented automatically and only areas containing worms, and their coordinates within the image, were stored. This procedure allowed a data reduction by approximately 1000 fold. For imaging, a copper frame (5.3 x 3.75 x 1 mm) was filled with 2% low melting point agarose (Sigma Aldrich) in M9 and 2–5 µl of a 10-fold concentrated overnight OP50 culture was seeded on top. The frame was deposited into a 10 cm NGM plate, and worms were transferred to the agarose arena. To preserve the moisture of the preparation and prevent shrinking of the gel, about ⅛ of the agar at the outer rim of the plate was removed using a scalpel and the space was filled with 6 ml of M9. Animals were recorded for at least 3 hr and all animals that were continuously tracked for at least 2 hr were included in the analyses in *Figure 5*.

### Dual bright-field and fluorescence imaging

Dual imaging was performed using an upright microscope (BX51WI; Olympus) and a 10 x objective (UplanSApo, NA 0.4; Olympus). For bright-field imaging, light emanating from a near-infrared (780 nm) LED (M780LP1 and driver LEDD1B; Thorlabs) was filtered using a (785/62 BrightLine HC; Semrock) and projected onto the sample via the bright-field illumination condenser. To excite fluorescence, the Teal line from an LED lamp (Spectra X light engine; Lumencor) was filtered (513/17 BrightLine HC; Semrock) and projected onto the sample using a 520 nm long-pass dichroic (FF520-Di02; Semrock). Transmitted and emitted light were filtered using a 532 long-pass filter (BLP01-532R; Semrock). To simultaneously record images in bright-field and fluorescence, a dual-camera device was used (DC²; Photometrics). Light was split into two channels using a 695 long-pass dichroic mirror (695DCXRUV; Photometrics) and images were projected into two cameras (acA3088-57um; BASLER). Fluorescent light was band-pass filtered (550/49 Brightline HC; Semrock) before reaching the camera sensor. The exposure time (6ms) of one camera served to synchronize the acquisition of the second camera and the Lumencor light engine. Individual worms were manually tracked using a 3-axis motorized stage (X-LSM150A; Zaber).

### Developmental pumping experiments

Worms were pre-synchronized by hypochlorite bleaching, allowed to hatch overnight in M9 and then cultured on NGM plates with OP50 at 20 °C. On day 3 after pre-synchronization, worms were synchronized again by letting 20 gravid animals lay eggs for 2 hr per assay plate, then removing the adults and letting embryos grow for specific durations to reach the appropriate larval or adult stage (19 hr for mid-L1, 31 hr for mid-L2, 39 hr for mid-L3, 50 hr for mid-L4, 65 hr for young adults). For the assay plates, 40 µl of *E. coli* OP50 culture was spotted onto an empty 6 cm NGM plate a few hours before the synchronization and left to dry. Synchronized worms were imaged directly on their assay plates as described in section 'Imaging setup'. The magnification for each stage was chosen to achieve an approximate pharynx length of ~60 pixels (2 x (1.18 µm/px) for L1, 1.5 x (1.57 µm/px) for L2, 1.4 x (1.69 µm/px) for L3, 1.3 x (1.81 µm/px) for L4 and the standard 1 x (2.36 µm/px) for young adults). Three assay plates were imaged once per stage, and three additional plates were imaged at each stage to test for photo-sensitivity.

### Starvation experiments

Washed animals were transferred to the center of an empty 6 cm NGM plate at room temperature and left to recover for 15 min before imaging. The same plate was imaged at defined time points for progressively more starved animals (at 30 min, 90 min, 150 min and 210 min after being taken off

food). The field of view was chosen randomly on the plate but was required to contain at minimum 3 worms at the beginning of the recording. For control, washed animals were transferred close to a 40 μl of *E. coli* OP50 lawn, which was spotted onto an empty 6 cm NGM plate a few hours before the recordings and allowed to dry. Acclimation time and recording are similar for starved animals.

## Automated analysis

### Pharyngeal pumping - fluorescence data

Animals were tracked using our custom python analysis package *PharaGlow* which is freely available under a permissive GPL 3.0 license. In brief, PharaGlow runs a three-step analysis: 1. center of mass tracking and collision detection, 2. linking detected objects to trajectories and 3. extracting centerline, contour, width, and other parameters of the shape to allow extracting pharyngeal pumping events. Tracking uses the soft matter package (*Allan et al., 2019*). The code is fully modular and any existing tracking code could in principle be used for the first two steps provided the input data is formatted to PharaGlow standards. We provide example data and example jupyter notebooks to help users make use of our package both in personal computer and high-performance cluster settings. The resulting files contain the position, and the straightened images which are further processed to extract the behavioral measures as described in *Figure 1* and section '*Pharyngeal pumping - postprocessing*".

### Pharyngeal pumping - postprocessing

To obtain pumping traces from straightened animals, the inverted maximum of the dorso-ventral standard deviation of brightness is calculated for each straightened frame per animal (*Figure 1E*). This metric is sensitive to the opening of the pharyngeal lumen and terminal bulb contractions. Peaks in the resulting trace correspond to pumping events. Due to the animal motion, uneven illumination or defocusing can modify the baseline of the pumping metric. We correct for baseline fluctuations and spurious fluorescence changes by subtracting the background fluctuations using a rolling mean filter of 1 s (except for *eat-18* mutants, where we use 3 s otherwise the slow contractions were removed too). To the remaining signal we apply a smoothing filter of width = 66 ms (2 frames). We detect peaks using AMPD, an algorithm for peak detection in quasi-periodic signals (*Scholkmann et al., 2012*). We also require the peak distances to obey physiologically reasonable rates i.e., the peaks can not be closer than $dmin = 132$ ms (4 frames). To automatically establish the noise level of the trace, we compare the incidence of intervals between detected peaks that violate the assumption $dmin > 132$ ms and select the minimal prominence required, such that the fraction of violating intervals is lower than a sensitivity parameter $s$. For all dataset with 5-min recordings, we set $s = 0.999$.

In the long-term recordings, we use a hampel filter with a width of 300 frames to remove spurious outliers in the signal which affect peak detection and set $s = 0.9999$.

Depending on the purpose, pumping rates have been calculated as follows: To determine the average pumping rate per track, we calculate the number of pumping events/ total track duration (e.g. *Figure 3F*, box plots). To obtain pumping rate distributions, we calculate the number of pumps in a sliding window of 10 s and combine data from all tracks. The 'instantaneous pumping rate' is defined as $1/\Delta t$ between pumps. We use the instantaneous rate when a higher temporal resolution is desired. Which rate metric is used is indicated in the caption.

### Assigning high and low pumping rate states

Pumping rates were calculated from the detected pumping events in a 30 s block window. The resulting rates were thresholded with a threshold of 2.5 Hz to discriminate between fast and slow pumping. The binary heatmap was filtered with a rolling median filter of width 5 min, which removes spurious events and allows extraction of longer term dynamics.

### Autocorrelation of pumping rates

The autocorrelation of the pumping rates for the 2 hr recordings was calculated from the 10 s average pumping rates (see *Pharyngeal pumping - postprocessing*). The decay time of the autocorrelation was determined using a one-sided t-test for each timepoint and calculating if the sample mean of the autocorrelation for each animal differed from 0. To determine the uncertainty of the decay time, we ran leave-one-out bootstrapping and report the mean and s.t.d. of the leave-one-out testing.

### Other behavioral parameters

Velocity was calculated from the tracked center of mass of the labeled pharynx. Reversals were calculated based on the angle between the pharynx and the animal's nose tip direction. To avoid spurious reversals, the nose tip trajectories are coarse-grained to 6 Hz, and the angle between the nose tip and pharynx is smoothed with a window of width = 1 s (30 frames). Timepoints with angles exceeding 120° were annotated as reversals. Reversals shorter than 0.5 s are removed. The estimation of the pharyngeal area is based on an automated threshold of the pharynx.

### Animal selection

All animals that were successfully tracked for at least 60 s (*Figures 1–3*) were included. No other filtering or outlier removal was performed. Due to age synchronization, all animals in the field of view were of similar size in the wildtype experiments. For *eat-18* mutants, the size and developmental stage of the animals were more dispersed and only animals that had the appropriate size for their stage were included (*Figure 5*). In the starvation experiments, animals that were successfully tracked for at least 20 s were included due to the larger velocity in this condition (*Figure 3*).

## Manual annotation of pumping behavior

Movies for individual animals were created from a large field of view and expert annotators counted pumps by displaying the movie using the cell counter tool in Fiji (*Schindelin et al., 2012*). The annotators were blinded to the movie conditions and to the other experts' results.

## Data and code availability statement

The data from this manuscript is available at https://osf.io/fy4ed/. The code repository for the PharaGlow package can be found at https://github.com/scholz-lab/PharaGlow (*Scholz, 2022*). Imaging software can be found at https://github.com/scholz-lab/Acquisition_PharaGlow, (*Bonnard, 2022* copy archived at swh:1:rev:29ae8971f66fa0722f4914deb4b70b7d53f76f07).

## Acknowledgements

We thank James Lightfoot and Shawn Lockery for helpful comments. We wish to acknowledge Omar Valerio Minero (MPINB HPC) for assistance with benchmarking. We thank Marc Pilon for giving us a plasmid. Some strains were provided by the CGC, which is funded by NIH Office of Research Infrastructure Programs (P40 OD010440). Some figure panels were created with biorender.com.

## Additional information

### Funding

| Funder | Grant reference number | Author |
| --- | --- | --- |
| Max Planck institute for Neurobiology of Behavior — caesar | Open access funding | Monika Scholz |

The funders had no role in study design, data collection and interpretation, or the decision to submit the work for publication.

### Author contributions

Elsa Bonnard, Data curation, Investigation, Methodology, Writing – review and editing; Jun Liu, Resources, Data curation, Investigation, Writing – review and editing; Nicolina Zjacic, Data curation, Investigation, Writing – review and editing; Luis Alvarez, Resources, Investigation, Writing – review and editing, Software; Monika Scholz, Conceptualization, Software, Funding acquisition, Methodology, Writing - original draft, Project administration, Writing – review and editing

### Author ORCIDs

Elsa Bonnard  http://orcid.org/0000-0001-6363-1684
Jun Liu  http://orcid.org/0000-0001-6472-3871

Nicolina Zjacic http://orcid.org/0000-0002-6413-7386
Luis Alvarez http://orcid.org/0000-0002-1027-2291
Monika Scholz http://orcid.org/0000-0003-2186-410X

**Decision letter and Author response**
Decision letter https://doi.org/10.7554/eLife.77252.sa1
Author response https://doi.org/10.7554/eLife.77252.sa2

## Additional files

### Supplementary files
• Transparent reporting form

### Data availability
The data from this manuscript is available at https://osf.io/fy4ed/. The code repository for the Phara-Glow package can be found at https://github.com/scholz-lab/PharaGlow.

The following dataset was generated:

| Author(s) | Year | Dataset title | Dataset URL | Database and Identifier |
|---|---|---|---|---|
| Scholz M, Bonnard E, Liu J | 2022 | Automatically tracking feeding behavior in populations of foraging worms | https://osf.io/fy4ed/ | Open Science Framework, fy4ed |

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

# Appendix 1

## Collision detection and effective number of independent animals in the field-of-view

PharaGlow detects all fluorescent objects in the field of view. The experimenter should choose a minimal and maximal size to allow only tracking objects that are single worms. If animals touch, an object that is larger than the maximally allowed size (maxSize) will be detected. In this case, the program automatically re-segments the region and attempts to separate the large object into multiple smaller objects using repeated filtering and thresholding. If this process is successful and two or more objects of the correct size can be separated, the program will continue tracking these animals. This approach is successful when animals touch, but do not overlap. We are unable to resolve crossings where animals are physically overlapping.

To determine the effect of collisions on tracking, we determine the number of detected objects in the field-of-view and compare it to the number of tracks obtained. Analysis of the track duration shows that average tracks are two min, and this duration depends on the velocity (*Figure 2—figure supplement 1*). We find that these two measures correlate well, supporting the view that animal tracks are not frequently broken into small tracklets (*Figure 2—figure supplement 1B*).

## Computational cost and scaling

We benchmarked the performance and scaling of the software using perfplot. *Appendix 1—table 1* details the pure computation time as run on a 1000 frame demo recording (available for download at the data repository). We find that parallelization improves performance for the object detection for up to 8 workers, and continues improving for >16 workers in the segmentation step. In our implementation, this option is already provided based on the python package multiprocessing. As the different steps depend on the details of the imaging, we have decided to report processing time per step.

1. Object detection
   The object detection step uses a full frame and does masking and object detection for each individual object. For this case, computational cost scales with the number of frames and can be easily parallelized to enable a faster speed. The average single-core computation time per frame is 300ms, which includes I/O, as we employ lazy data loading, which allows analyses of data that are much larger than the RAM available. Of note, this step can be omitted if our acquisition software is used (see Methods) as single worms are segmented already during acquisition.
2. Tracking and trajectory interpolation
   The tracking step is based on trackpy and here the scaling depends on the search range (how far can an object move between frames) and the memory (how many subsequent frames can an object be unobserved). Typically, this step is much faster than the other two as it does not handle large I/O or image processing.
3. Segmentation, centerline detection, straightening, and pump detection

   In this step, the previously detected images of detected pharynx are further processed. The total compute time here depends on the product between the number of objects and the number of recording frames. We therefore provide a per-object assessment of the processing time.

**Appendix 1—table 1.** Benchmarking of the typical computing times.
Full frames are the multi-animal images with 3088x2064 px. Mini-frames are small regions of interest with one animal (*Figure 1D*).

| Step | Compute time* *Intel(R) Xeon(R) Gold 6230 CPU @ 2.10 GHz | | |
| --- | --- | --- | --- |
| | 1 worker | 4 workers | 16 workers |
| Object detection | 186.6 ms / full frame | 81.2 ms / full frame | 57.4 ms / full frame |
| Tracking and trajectory interpolation | 1 ms / miniframe | 1 ms / miniframe | 1 ms / miniframe |
| Segmentation etc. | 378 ms / mini-frame | 104 ms / miniframe | 32 ms / miniframe |

*Appendix 1—table 1 Continued on next page*

*Appendix 1—table 1 Continued*

| | Compute time*<br>*Intel(R) Xeon(R) Gold 6230 CPU @ 2.10 GHz | | |
|---|---|---|---|
| **Typical recording** 150 worm minutes of data (9000 full frames, 30 worms) | **1.2** days | **8.2** h | **3** h |

## Effects of light exposure

*C. elegans* are known to sense and react to light by initiating reversals and suppressing pumping. These reactions occur more frequently at short wavelengths and high power densities (*Ward et al., 2008*; *Bhatla et al., 2015*; *Bhatla and Horvitz, 2015*). To determine if our imaging conditions affected behavior, we measured the light intensity and the leaving rates of animals during imaging. We used excitation light centered at 500 nm, and measured an effective intensity of only 0.24 mW/mm$^2$ in the field-of-view, 54 times lower than the reported intensity that induces pumping inhibition or spitting[3]. We observed 5–25% of animals leaving the field-of-view during recordings, indicating a mild avoidance reaction which depends on the developmental stage (*Figure 2—figure supplement 3*).

To control for photo-toxic effects, we split our developmental cohort into two groups. One group of animals was imaged consecutively at each larval stage (multiple exposures), the other group was left to grow under the same conditions, but only ever imaged once (single exposure). We find that during all larval stages, the behavioral results of the two groups are similar, but not in young adults (*Figure 2—figure supplement 4*). For the young adult cohort, the animals that were repeatedly imaged show a higher velocity compared to the single-exposure group, as well as differences in all other behavioral metrics we report. We speculate that this could be due to differences in drying of the plates during repeated imaging, or a possible light-induced chronic effect.

Over longer time scales, exposure to light can reduce the worms' lifespan (*De Magalhaes Filho et al., 2018*). To test for chronic photo-toxic effects, we tested the viability of worms after long exposure to 500 nm light. We continuously illuminated 30 young adult GRU101 worms for five hours using the same illumination intensity as our PharaGlow assay (0.24 mW/mm$^2$ at the focal plane). We employed a copper frame to prevent the worms from escaping the illuminated area. A plate not exposed to the 500 nm light was placed on the same bench close to the microscope as our negative control. After five hours, the copper frame was removed and worms were scored for viability both immediately and after overnight recovery in a 20 °C incubator. All worms were viable and able to move upon gentle tapping on the plates immediately after illumination. Further checking after overnight recovery confirmed their continued viability. This suggests that exposure to this light level does not cause observable photo-toxic effects.

To further assess the impact of the excitation light on behavior, we measured on-food pumping in adult animals expressing the red fluorophore mCherry compared to the strain expressing YFP, which is used throughout the paper. If the impact of the wavelength is non-negligible, the red fluorophore mCherry (excitation centered at 587 nm) should result in fewer reversals or accelerations compared to YFP, as these responses are wave-length dependent (*Ward et al., 2008*; *Bhatla et al., 2015*; *Bhatla and Horvitz, 2015*). As expected, we find an increase in the velocity for the green light exposed animals (*Figure 2—figure supplement 5A*). However, we find that pumping rates between the two populations are not significantly different (*Figure 2—figure supplement 5B*), suggesting that these light intensities do not affect pumping behavior. We note that the differential effects of excitation light on behavior should be taken into account when investigating the coupling between e.g., locomotory and feeding behaviors.

