## [Editor Report]

In this study, Bonnard and colleagues report a new method to assay feeding rates in *C. elegans*. Imaging fluorescence in the pharynx with subsequent image processing steps they make it possible to record pharyngeal pumping across freely behaving animal populations over periods up to 3 hs. They validate their method in different behavioural paradigms and with various feeding mutants.

---

## [Decision Letter]

**Decision letter after peer review:**

[Editors’ note: the authors submitted for reconsideration following the decision after peer review. What follows is the decision letter after the first round of review.]

Thank you for submitting the paper "Automatically tracking feeding behavior in populations of foraging worms" for consideration by *eLife*. Your article has been reviewed by 3 peer reviewers, one of whom is a member of our Board of Reviewing Editors, and the evaluation has been overseen by a Senior Editor. The following individual involved in the review of your submission has agreed to reveal their identity: Serena Ding (Reviewer #2).

Comments to the Authors:

We are sorry to say that, after consultation with the reviewers, we have decided that this work will not be considered further for publication by *eLife*. The reviewers found your work of high quality, and upon some revisions could represent a useful technique to study feeding behaviors in worms. However, the consensus of the reviewers was that your new method is useful only for this small circle of *C. elegans* researchers and therefore does not address the broader audience of *eLife* readers.

Please find their detailed comments and recommendations below.

*Reviewer #1 (Recommendations for the authors):*

(1) I think the authors did not fully exploit the great potential to perform long term observations in individual worms. I wonder whether they could perform experiments on single or few worms restricted to the imaging field of view so that they cannot collide and escape; thus, single worm identity is kept. How long is it possible to record without bleaching the pharyngeal marker? Is it possible to record over extended episodes, like feeding quiescence during lethargus or upon stress induction?

(2) Related to the above. Pumping rates in general show spread distributions, which I believe are not due to noise, please clarify. And under some conditions distributions show interesting bi-modality or skew e.g. Figure 4B (30min starvation) or unc-31 mutants (Figure 5F). Are inter-pump intervals autocorrelated and change smoothly in fed WT animals, and if yes on which timescales? Do 30min starved animals and unc-31 mutants switch between discrete states of high/low pumping rates and if yes, on which timescales. The authors report that they observed episodes of feeding quiescence, perhaps related to satiety quiescence. They could be more quantitative about this observation describing the time-scales and frequencies of these observations.

(3) The authors should be clearer about what n-numbers / sample sizes represent. Individual assays, worms used in assays (which would be less than data in the field of view), tracklets (which can be more than observed worms, see methods: 20-30s tracklets, meaning that many track interruptions could artificially blow-up the n-numbers) … ? E.g., in Figure 3 n numbers are in the hundreds; in Figure 3E two numbers are given in brackest. Please indicate these unambiguously in the Figure captions.

(3a) Depending on this, n numbers might be inaccurate in describing the real number of datapoints; or datapoints might not be independent of each other (e.g. many tracklets from the same worm). This affects the right choice of statistical testing procedures. Most figure panels that show comparisons lack any statistics.

(4) The authors state that a simple laptop/desktop would be sufficient but it seemed they used an HPC to generate the data in the manuscript. Please be clear about what the requirements really are and if the analyses are really feasible in a reasonable time using a standard computer. What were the CPU hours required to generate the data in this study?

(5) … "also find that similar to off-food reversals, which are constant throughout

larval development"… prior to this section the authors should explain better the general *C. elegans* behaviours known as some readers might not know what reversals are etc. For them, this comes out of nowhere.

(6) Figure 5D: the different contraction patterns should be qualitatively better described in the text so that readers get a better intuition of what causes the differences in the mean change images.

(7) The authors discuss a future outlook applying their method to 3D behaviour. For me, it is hard to imagine how this could ever work. Be precise, how this could be achieved realistically or remove this statement.

(8) …" using a rolling mean filter of 1 s and a smoothing filter of width =

66 ms (2 frames)": please be precise what the smoothing filter was, and why was this needed, since there is already a 1s rolling mean applied.?

*Reviewer #2 (Recommendations for the authors):*

– This new method directly measures the pumping of the pharyngeal muscles as a proxy for feeding. However, worms are also known to pump their pharynx without actual food intake, such as when off-food. The authors should be more explicit about this limitation to their method and take care when using "pumping" and "feeding" interchangeably.

– One key advantage of this new method is that pumping and locomotion behaviour can be simultaneously detected to generate new insights e.g. regarding behavioural modularity and coordination. However, the authors report that while the pumping rates remain the same whether imaging with YFP or mCherry, other locomotion metrics such as velocity are different. This result thus calls for careful interpretation in future studies that link both types of behaviours using this method.

– The first sentence of the introduction (p.1) "Animals must forage… and provide for their young" is perhaps too generalised. There are animals that do not provide for their young, so I suggest removing this part of the sentence.

– Page 8, middle paragraph: multiple exposure vs. single exposure experimental differences "could be due to … different remaining food levels". Why would this be, if the experiments and feeding rates are reported to be the same in both cases?

– "Pharaglow" is capitalised in some cases and not others.

– Perhaps the authors could also comment on what happens when animals overlap during the experiment, especially in the context of large scale foraging experiments.

*Reviewer #3 (Recommendations for the authors):*

Detailed comments:

The term "worm" is highly nonspecific, encompassing 3 phyla.

"*C. elegans*" or better yet "*Caenorhabditis elegans*" should be in the title.

The Introduction focuses somewhat narrowly on studies of foraging strategies and food intake. In addition to these interesting topics, the authors could discuss how assessment of feeding is used in studies of behavioral genetics, quiescence, and aging.

I found the description of pharyngeal pumping in the introduction to be quite confusing. The authors should strive to use standard anatomical and behavioral terms and explain them for the naive reader. For example, the sentence "Transport proceeds with occasional peristaltic contractions that move food further toward the intestine where a hard cuticular structure, the grinder, crushes the bacteria before they are pushed into the intestine" needs to be rephrased. The peristaltic motion is called isthmus peristalsis. It moves food particles to the terminal bulb which contains the grinder.

"Of these motions, pumping is the most frequent contraction that is also the limiting step for food intake". I am lost as to what was intended here.

Figure 1F: Why is the y axis given as arbitrary units? I would like to see how large the variation in standard deviation is during pharyngeal pumping.

The comparison between automated assessment and expert manual assessment is weak because both were done with low-resolution fluorescence data. If the goal is to compare with how pharyngeal pumping is normally assessed, the authors should use high-resolution bright field or DIC images, as

Another problem with the automated/manual comparison is that the pumping rate varied over a fairly narrow range. As the authors acknowledge, pumping is modulated by many different factors, so it is not difficult to prepare worms with a wide range of pumping rates. Doing this would help assess linearity of the method, especially at low pump rates.

The sections describing various applications of the method (development, starvation, and mutants) are poorly motivated and written in a confusing manner. For each section the authors need to briefly introduce the questions being addressed and why they are important. Similarly, the results should be briefly discussed in light of the questions being asked.

Page 7: "We find that on-food pumping rates increase slightly over the course of the larval stages, but much less dramatically than the velocity." Why are the authors comparing the increase in pumping rate with velocity? These are like apples and oranges.

The authors cite results showing that long-term exposure to light reduces lifespan. But to test for phototoxicity, the authors measure not lifespan but feeding rate! This seems odd. A simple experiment would be to keep the illumination on until all the animals have died. This would give some indication of how close to a lethal dose is being delivered to the worms.

Figure 3: The ordering of the panels here is confusing.

3C,3D: Specific Y axis labels would be better here. Why not measure area in µm^2? Why not have axes for pumping rate in Hz and velocity in mm/s?

Page 10: "…the distributions of both pumping rates and velocities show distinct sub-populations". I see no evidence of subpopulations in velocity. There are some local maxima and minima in the pumping rate, but without further analyses these effects seem quite preliminary.

Page 12. "It is possible that some of the detected pumps in our measurements are either peristaltic movements or other non-productive muscular motion". The authors imply that isthmus peristalsis is "non-productive", which does not make sense to me. Have the authors observed isthmus peristalsis in their images? It should be straightforward to see if isthmus peristalsis is reflected in the traces shown in Figure 1F.

Page 16: "labeling parts of the pharynx with lipophilic dyes would be a possibility to extend the usage of this tool beyond species that are genetically tractable". I do not understand how lipophilic dyes, which label certain structures but not the pharynx, would be useful here.

[Editors’ note: further revisions were suggested prior to acceptance, as described below.]

Thank you for resubmitting your work entitled "Automatically tracking feeding behavior in populations of foraging worms" for further consideration by *eLife*. Your revised article has been evaluated by Ronald Calabrese (Senior Editor) and a Reviewing Editor. I am happy to tell you that all reviewers are now very enthusiastic about your manuscript, which has been significantly improved. However, there are some remaining issues raised by reviewer #1 that need to be addressed, as outlined below.

*Reviewer #1 (Recommendations for the authors):*

In this revision, the authors have made some efforts to address my review points. My main point, however, which was to prove the suitability of their approach for long term observations with subsequent analyses is only partially addressed.

(1) It is very promising to see that there is almost no bleaching of the pharyngeal signal; this does not mean however that these experiments work practically for extended recording times of more than one hour (see #2 below). I suggest being more careful with the conclusions.

(2) The authors show an impressive example of male-hermaphrodite interactions recording behaviors over a period of 1h. This is great, but unfortunately just one example. Is this the best outcome the authors could have ever achieved or is it representative for many experiments? With the male-hermaphrodite paradigm, the authors go beyond what I was requesting. If their approach works as they claim, it should be feasible to perform a sufficient amount of single animal recordings.

(3) The authors attempted to address one of my points showing pumping rate distributions of individual animals (2H, 3G). I disagree with the authors statement that PharaGlow can "produce large animal statistics while preserving single worm behavioral information". This ability is hampered by the short tracklet durations. One cannot conclude from e.g. 3G left panel whether individuals differ in an idiosyncratic way versus transient changes that are randomly captured in the short tracklet episodes. Hence, my previous request to perform statistics on longer recordings. Figure 5D indicates that the animals exhibit minutes lasting episodes of high and low pumping rates. More 1h recordings on individuals like in Figure 5 (perhaps no need to do male plus hermaphrodite) will enable the authors to perform the requested analyses. As mentioned above, if PharaGlow performs as the authors claim, these revisions should be doable with reasonable effort.

*Reviewer #2 (Recommendations for the authors):*

The authors have thoroughly and effectively responded to previous reviews, resulting in a significantly improved manuscript. It is now clear that the method is broadly applicable using common imaging microscopes and computing resources, and that it is suitable for studying pumping behaviour in populations of unrestrained animals across various time- and space- scales. The method affords novel observations not directly feasible with previous methods, for instance, exemplified by the authors' recent addition of pumping dynamics during mating events.

*Reviewer #3 (Recommendations for the authors):*

The statistical tests added to the paper were not reported correctly. There are multiple instances of "p<0.000" and ""p=0.000" in the figures and captions. In fact every single p value given in Figure 4 and its caption is either zero or negative! Similarly in Figure S2.2A , S2.5A.

But in general the revised manuscript is much improved compared to the original.

---

## [Author Response]

[Editors’ note: The authors appealed the original decision. What follows is the authors’ response to the first round of review.]

Reviewer #1 (Recommendations for the authors):(1) I think the authors did not fully exploit the great potential to perform long term observations in individual worms. I wonder whether they could perform experiments on single or few worms restricted to the imaging field of view so that they cannot collide and escape; thus, single worm identity is kept. How long is it possible to record without bleaching the pharyngeal marker? Is it possible to record over extended episodes, like feeding quiescence during lethargus or upon stress induction?

We thank the reviewer for this important suggestion, which we now added to the manuscript. Due to the large data rate of these recordings, we combined a live-segmentation with our down-stream analysis. The results are now presented as Figure 5 of the revised manuscript.

Furthermore, we provide additional data characterizing photo-bleaching. This data shows that continuous recordings for extended periods of time (> 5 hrs) are feasible using this approach (Figure S5.1).

(2) Related to the above. Pumping rates in general show spread distributions, which I believe are not due to noise, please clarify. And under some conditions distributions show interesting bi-modality or skew e.g. Figure 4B (30min starvation) or unc-31 mutants (Figure 5F). Are inter-pump intervals autocorrelated and change smoothly in fed WT animals, and if yes on which timescales?

We thank the reviewer for bringing up an important point regarding the stochasticity observed in pumping, and we discuss this now in more detail in the introduction. Off-food and low food pumping rates show interesting inter-pump dynamics (see e.g., Scholz et al., *PNAS* 2017) and also Figure 4 of this manuscript. In contrast, the on-food behavior shows mostly steady pumping at constant rates, as can be seen from the on-food pumping rate distribution in Figure S3.1. Broader distributions in on-food conditions arise from inter-animal variability in the peak rates and brief pauses in pumping, or in the case of *unc-31* due to effects of the mutant background. To illustrate individual animal data, we have added histograms for individual animals off-food as well as individual animal timeseries to Figures 2 and 3. We now introduce previous work highlighting the inherent stochasticity of pumping. Additionally, we describe in more detail in the methods section how we calculate pumping rates, as there is some variation depending if average pumps/ unit time or instantaneous rates are reported.

Do 30min starved animals and unc-31 mutants switch between discrete states of high/low pumping rates and if yes, on which timescales.

The reviewer raises an interesting question. To examine if there are switches between high and low pumping rates, we now present the pumping rate distributions for individual animals. For our experiments, animals appear to show distinct mean rates and not switch on the timescale of minutes (Figures 2H and 3G).

Understanding the timescale on which switches would occur and how this might differ in mutant animals would make for an interesting follow up study when combined with our long-term imaging (Figure 5).

The authors report that they observed episodes of feeding quiescence, perhaps related to satiety quiescence. They could be more quantitative about this observation describing the time-scales and frequencies of these observations.

As the reviewer recommends, we have added a quantification of the observed non-pumpers (<1% for most recordings). Interestingly, we also observe long feeding quiescence in Figure 5 (male), indicating regulation at the tens of minutes timescale.

(3) The authors should be clearer about what n-numbers / sample sizes represent. Individual assays, worms used in assays (which would be less than data in the field of view), tracklets (which can be more than observed worms, see methods: 20-30s tracklets, meaning that many track interruptions could artificially blow-up the n-numbers) … ? E.g., in Figure 3 n numbers are in the hundreds; in Figure 3E two numbers are given in brackest. Please indicate these unambiguously in the Figure captions.

We thank the reviewer for pointing out this potentially confusing wording. We now state that the N numbers indicate animal tracks when referring to automatically analyzed data and have clarified the positioning and description of our N in the figures and captions. We would like to re-emphasize that we indeed record hundreds of animals in the experiments described in Figure 3. We record over 30 individuals in one field of view, and repeat this for 6 independent plates. We believe the large sample size is a major strength of our method and have altered the text to reflect this.

We have additionally added a section to the supplementary information showing the average track duration and how it relates to the number of individuals we record (Figure S3.6). As we state in the text, we only consider tracklets that are at least 1 min, except for the starvation data. However, this does not mean we only track for such short periods of time. Instead, our average track length is ~ 2 min (5 min recording time) and scales with velocity, indicating that leaving the field-of-view, rather than crossings, is what limits the tracking duration (Figure S2.1). We also describe how the tracker resolves crossings, which we are successful in doing except when worms are physically overlapping.

(3a) Depending on this, n numbers might be inaccurate in describing the real number of datapoints; or datapoints might not be independent of each other (e.g. many tracklets from the same worm). This affects the right choice of statistical testing procedures. Most figure panels that show comparisons lack any statistics.

Given the sample size and the additional analyses of track-length and track numbers as detailed above, we are confident that we are not vastly over-estimating our sample size and that our chosen tests are appropriate. We have added p-values to all pertinent figures (3,4,5 and supplementary).

(4) The authors state that a simple laptop/desktop would be sufficient but it seemed they used an HPC to generate the data in the manuscript. Please be clear about what the requirements really are and if the analyses are really feasible in a reasonable time using a standard computer. What were the CPU hours required to generate the data in this study?

We appreciate the reviewer’s suggestions regarding additional analyses of the required compute times. We have added a benchmarking of the PharaGlow analysis in the supplementary information, showing expected times for both an end user laptop (single core and 4 core), as well as on a typical HPC. Data analysis, as we state in the manuscript, is possible on a normal laptop, and one can expect analysis to take around ~ 8 h for a 4-fold parallelization on CPU for a typical dataset presented in this paper of 3 worm*hours of data (see Table TS1 for comparison and performance in different scenarios).

However, due to the total scale of our datasets we have chosen to use a central storage and compute cluster. Our benchmarking shows that parallelization is very effective.

(5) … "also find that similar to off-food reversals, which are constant throughoutlarval development"… prior to this section the authors should explain better the general *C. elegans* behaviours known as some readers might not know what reversals are etc. For them, this comes out of nowhere.

We thank the reviewer for this comment. In response, we have expanded our discussion of locomotory behaviors and provided more context for the reversal measurements. We now include reversal rates in Figure 3, as well as the rescaled measurements to match other studies’ measurements in Figure S2.2. We have also included the following description in the text: “Animals move forward on agar by generating waves of muscular contraction through their body. When the animals reverse the direction of these waves, they move backwards”

(6) Figure 5D: the different contraction patterns should be qualitatively better described in the text so that readers get a better intuition of what causes the differences in the mean change images.

We appreciate the reviewer's suggestion to describe the contraction patterns. We discuss this in the text and now show the average peak shape in Figure 4E and S4.1 which highlights the slower, longer duration contractions in the *eat-18* mutant.

(7) The authors discuss a future outlook applying their method to 3D behaviour. For me, it is hard to imagine how this could ever work. Be precise, how this could be achieved realistically or remove this statement.

We envisioned using fast 3D imaging, e.g., using multifocal imaging, which could in principle enable such measurements at the required rates. However, to keep the discussion focused and considering our emphasis on simple imaging techniques, we have removed this statement.

(8) …" using a rolling mean filter of 1 s and a smoothing filter of width = 66 ms (2 frames)": please be precise what the smoothing filter was, and why was this needed, since there is already a 1s rolling mean applied.?

We thank the reviewer for pointing out the misleading wording of this sentence. It now reads: *We correct for baseline fluctuations and spurious fluorescence changes by subtracting the background fluctuations using a rolling mean filter of 1 s. To the remaining signal we apply a smoothing filter of width = 66 ms (2 frames).*

Reviewer #2 (Recommendations for the authors):– This new method directly measures the pumping of the pharyngeal muscles as a proxy for feeding. However, worms are also known to pump their pharynx without actual food intake, such as when off-food. The authors should be more explicit about this limitation to their method and take care when using "pumping" and "feeding" interchangeably.

We thank the reviewer for bringing up this important point of nomenclature, and have corrected the manuscript to read “pumping” throughout when we refer to our measurements.

– One key advantage of this new method is that pumping and locomotion behaviour can be simultaneously detected to generate new insights e.g. regarding behavioural modularity and coordination. However, the authors report that while the pumping rates remain the same whether imaging with YFP or mCherry, other locomotion metrics such as velocity are different. This result thus calls for careful interpretation in future studies that link both types of behaviours using this method.

We appreciate the reviewer highlighting this differential effect of light on different behaviors. The light effects on locomotor behaviors are well established, and we agree that these have to be carefully controlled. To emphasize this in our manuscript we now state: *We note that the differential effects of excitation light on behavior should be taken into account when investigating the coupling between e.g., locomotory and feeding behaviors.*

– The first sentence of the introduction (p.1) "Animals must forage… and provide for their young" is perhaps too generalised. There are animals that do not provide for their young, so I suggest removing this part of the sentence.

We thank the reviewer for this suggestion, and have removed this statement as suggested.

– Page 8, middle paragraph: multiple exposure vs. single exposure experimental differences "could be due to … different remaining food levels". Why would this be, if the experiments and feeding rates are reported to be the same in both cases?

We thank the reviewer for pointing this out, we have removed this statement.

– "Pharaglow" is capitalised in some cases and not others.

We thank the reviewer for pointing out the inconsistency, we have now formatted all instances to on spelling.

– Perhaps the authors could also comment on what happens when animals overlap during the experiment, especially in the context of large scale foraging experiments.

We thank the reviewer for this suggestion. We have added a section to the supplementary information demonstrating how our software deals with collisions and overlaps, and how this impacts tracking. In brief, we employ collision detection and attempt to resolve objects where two or more animals touch. This is successful when animals touch, however we can not yet resolve overlapping animals (and the pumping signals obtained from these animals would be uninterpretable).

Reviewer #3 (Recommendations for the authors):Detailed comments:The term "worm" is highly nonspecific, encompassing 3 phyla."*C. elegans*" or better yet "*Caenorhabditis elegans*" should be in the title.

We agree with the reviewer and have changed the title to specify *C. elegans*.

The Introduction focuses somewhat narrowly on studies of foraging strategies and food intake. In addition to these interesting topics, the authors could discuss how assessment of feeding is used in studies of behavioral genetics, quiescence, and aging.

We thank the reviewer for this helpful suggestion. In the revised version we extend the introduction and include references to pertinent studies across subfields, including behavioral genetics, quiescence, aging, and pharmacological screens, that use feeding behavior as a read-out.

I found the description of pharyngeal pumping in the introduction to be quite confusing. The authors should strive to use standard anatomical and behavioral terms and explain them for the naive reader. For example, the sentence "Transport proceeds with occasional peristaltic contractions that move food further toward the intestine where a hard cuticular structure, the grinder, crushes the bacteria before they are pushed into the intestine" needs to be rephrased. The peristaltic motion is called isthmus peristalsis. It moves food particles to the terminal bulb which contains the grinder.

We thank the reviewer for pointing out this editing mistake. The corrected sentence reads as follows: ‘Transport of the bacteria proceeds with occasional peristaltic contractions that move food further toward the terminal bulb, where a hard cuticular structure, the grinder, crushes the bacteria before they are pushed into the intestine._’_

"Of these motions, pumping is the most frequent contraction that is also the limiting step for food intake". I am lost as to what was intended here.

In light of this comment and a related comment by reviewer 2, we have expanded our discussion of the terms feeding and pumping as used in the field.

Figure 1F: Why is the y axis given as arbitrary units? I would like to see how large the variation in standard deviation is during pharyngeal pumping.

We thank the reviewer for pointing this out. In response to the reviewer’s suggestion we have added a scale bar to the y-axis. We would like to point out that the D-V variation scales with the average brightness and its absolute magnitude can therefore vary depending on expression level, camera settings and excitation light.

The comparison between automated assessment and expert manual assessment is weak because both were done with low-resolution fluorescence data. If the goal is to compare with how pharyngeal pumping is normally assessed, the authors should use high-resolution bright field or DIC images, as

We would like to emphasize that resolution, rather than magnification, is relevant here. We report manual counts in Figure 2 of the original submission at a pixel sizeof 1.2 µm/px and a resolution, based on the Rayleigh criterion of ~1.3 µm, which easily resolves the opening of the lumen of the corpus. However, as we state in response to the next comment, we have provided additional data at higher magnification (10x) for a range of pumping rates to demonstrate the accuracy of the method (see new Video2).

Another problem with the automated/manual comparison is that the pumping rate varied over a fairly narrow range. As the authors acknowledge, pumping is modulated by many different factors, so it is not difficult to prepare worms with a wide range of pumping rates. Doing this would help assess linearity of the method, especially at low pump rates.

We thank the reviewer for this suggestion to help strengthen our conclusion by additional verification of the method at a larger range of pumping rates than what was shown in Figure 2 and Figure S3.4. We now provide additional support for the accuracy of our method by showing data from simultaneous bright-field and fluorescence recordings of worms in a wide range of conditions to elicit different pumping rates. We find that Pharaglow accurately detects pumps across the relevant physiological range. These data are now shown as Figure 1H-J.

The sections describing various applications of the method (development, starvation, and mutants) are poorly motivated and written in a confusing manner. For each section the authors need to briefly introduce the questions being addressed and why they are important. Similarly, the results should be briefly discussed in light of the questions being asked.

We appreciate this suggestion by the reviewer to improve the readability of our manuscript. We have revised these sections accordingly.

Page 7: "We find that on-food pumping rates increase slightly over the course of the larval stages, but much less dramatically than the velocity." Why are the authors comparing the increase in pumping rate with velocity? These are like apples and oranges.

In response to the reviewer's comment we have rephrased this sentence to read:

We find that on-food pumping rates increase slightly over the course of the larval stages, but much less dramatically than the velocity increases over development, despite the substantial growth of both the body and the pharyngeal muscle.

We hope this clarifies our intent to emphasize that some behaviors are altered by the changes and added muscle mass during development, but others are unaffected.

The authors cite results showing that long-term exposure to light reduces lifespan. But to test for phototoxicity, the authors measure not lifespan but feeding rate! This seems odd. A simple experiment would be to keep the illumination on until all the animals have died. This would give some indication of how close to a lethal dose is being delivered to the worms.

We appreciate this suggestion. In response to this question, we have re-organized the results section to separate acute and long-term photo-toxic effects. We have employed a long-term exposure of five hours continuous illumination and found that longer exposure to this level of light does not cause life-threatening phototoxic effects.

We thank the reviewer for their comments and suggestions about a life span experiment. It would be interesting to see the effect of long exposure but it is outside the current scope of our manuscript. We have demonstrated in Figures 3 and 5 that normal/extended exposure times of light have no impact on measured phenomena.

Figure 3: The ordering of the panels here is confusing.

We thank the reviewer for this point. We have revised Figure 3 with a clearer layout.

3C,3D: Specific Y axis labels would be better here. Why not measure area in µm^2? Why not have axes for pumping rate in Hz and velocity in mm/s?

We would like to point out that the data with appropriate units are present in Figure 3, as the box plot and the normalized panels are from the same dataset, but these two visualizations serve different purposes. We find that the normalized data are useful to compare rates with the adult stage, which is behaviorally most well characterized.

We have also added the area measurements in um^2 to the supplementary information in Figure S3.1.

Page 10: "…the distributions of both pumping rates and velocities show distinct sub-populations". I see no evidence of subpopulations in velocity. There are some local maxima and minima in the pumping rate, but without further analyses these effects seem quite preliminary.

We thank the reviewer for pointing out this inaccuracy in our language. We have revised the sentence and corresponding figure reference to clearly indicate the joint distribution in panel 4F, and have added additional panels demonstrating the sub-populations more convincingly using single-animal data.

Page 12. "It is possible that some of the detected pumps in our measurements are either peristaltic movements or other non-productive muscular motion". The authors imply that isthmus peristalsis is "non-productive", which does not make sense to me. Have the authors observed isthmus peristalsis in their images? It should be straightforward to see if isthmus peristalsis is reflected in the traces shown in Figure 1F.

We have not investigated the presence of isthmus peristalsis in our images and this would be beyond the scope and goal of this paper. We have revised the corresponding section to avoid appraising the productive value of peristalsis. We thank the reviewer for pointing this out.

Page 16: "labeling parts of the pharynx with lipophilic dyes would be a possibility to extend the usage of this tool beyond species that are genetically tractable". I do not understand how lipophilic dyes, which label certain structures but not the pharynx, would be useful here.

We have removed this sentence from the discussion.

[Editors’ note: what follows is the authors’ response to the second round of review.]

Reviewer #1 (Recommendations for the authors):In this revision, the authors have made some efforts to address my review points. My main point, however, which was to prove the suitability of their approach for long term observations with subsequent analyses is only partially addressed.(1) It is very promising to see that there is almost no bleaching of the pharyngeal signal; this does not mean however that these experiments work practically for extended recording times of more than one hour (see #2 below). I suggest being more careful with the conclusions.

We appreciate the reviewer's comments regarding the interpretation of the bleaching curves. We have since added a larger number of longer recordings (3h) to support our statements (see responses below).

(2) The authors show an impressive example of male-hermaphrodite interactions recording behaviors over a period of 1h. This is great, but unfortunately just one example. Is this the best outcome the authors could have ever achieved or is it representative for many experiments? With the male-hermaphrodite paradigm, the authors go beyond what I was requesting. If their approach works as they claim, it should be feasible to perform a sufficient amount of single animal recordings.

We agree with the reviewer and have added a substantial number of long-term recordings to this revised manuscript (a total of 37 recordings over 3 genotypes, recording time of 3 h, we selected tracks of > 2 h duration). We wanted this data to be meaningful beyond observing wild type behavior, and have therefore decided to also assay the long-term pumping behavior of the *unc-31* mutants. We believe these data demonstrate the feasibility of the method for at least 2h long recordings. New data is provided on panels 5G-J.

(3) The authors attempted to address one of my points showing pumping rate distributions of individual animals (2H, 3G). I disagree with the authors statement that PharaGlow can "produce large animal statistics while preserving single worm behavioral information". This ability is hampered by the short tracklet durations. One cannot conclude from e.g. 3G left panel whether individuals differ in an idiosyncratic way versus transient changes that are randomly captured in the short tracklet episodes. Hence, my previous request to perform statistics on longer recordings. Figure 5D indicates that the animals exhibit minutes lasting episodes of high and low pumping rates. More 1h recordings on individuals like in Figure 5 (perhaps no need to do male plus hermaphrodite) will enable the authors to perform the requested analyses. As mentioned above, if PharaGlow performs as the authors claim, these revisions should be doable with reasonable effort.

We thank the reviewer for pointing out this interesting aspect of animal behavioral dynamics. Based on the fact that we use isogenic, age-synchronized animals with the same life-history, we could expect that the short-term sampling of many animals corresponds to long-term sampling of one animal (assumption of ergodicity). We agree however, that given the animal individuality reported in e.g., *C. elegans* and *D. melanogaster*, this assumption likely does not fully hold. To address the question whether there are state switches in animal behavior as seen for the mating pair in Figure 5 and as suggested by the statistics shown in Figure 3, we have performed long-term recordings of wt and *unc-31* mutant animals. We analyzed tracks that are at least 2 hours long. We observe that *unc-31* animals switch between a high/low pumping state less frequently than wildtype. Additionally, we now state “Taken together, these results show that studying the underlying behaviors and dynamics in a worm population requires large statistics and long recordings. Depending on the desired data, both long-term recordings and short-term high-throughput measurements are accessible with PharaGlow.” We appreciate the suggestion of the reviewer.

Reviewer #3 (Recommendations for the authors):The statistical tests added to the paper were not reported correctly. There are multiple instances of "p<0.000" and ""p=0.000" in the figures and captions. In fact every single p value given in Figure 4 and its caption is either zero or negative! Similarly in Figure S2.2A , S2.5A.But in general the revised manuscript is much improved compared to the original.

We thank the reviewer for detecting this issue. We have now reformatted the p-values to the expected format e.g., p < 0.001 or p < 0.0001 or all figures and captions. We apologize for the confusion.